# Opposing, spatially-determined epigenetic forces impose restrictions on stochastic olfactory receptor choice

Elizaveta V Bashkirova[1,2], Nell Klimpert[3], Kevin Monahan[4], Christine E Campbell[5,6], Jason Osinski[5,6], Longzhi Tan[7], Ira Schieren[2], Ariel Pourmorady[1,2], Beka Stecky[2], Gilad Barnea[3], Xiaoliang Sunney Xie[8,9], Ishmail Abdus-Saboor[2], Benjamin M Shykind[10], Bianca J Marlin[2], Richard M Gronostajski[5,6], Alexander Fleischmann[3], Stavros Lomvardas[2,11]*

[1]Integrated Program in Cellular, Molecular and Biomedical Studies, Vagelos College of Physicians and Surgeons, Columbia University Irving Medical Center, Columbia University, New York, United States; [2]Zuckerman Mind, Brain, and Behavior Institute, Columbia University, New York, United States; [3]Department of Neuroscience, Division of Biology and Medicine and Robert J. and Nancy D. Carney Institute for Brain Science, Brown University, Providence, United States; [4]Department of Biochemistry and Molecular Biology, Rutgers University, Newark, United States; [5]Department of Biochemistry, University at Buffalo and New York State Center of Excellence in Bioinformatics and Life Sciences, Buffalo, United States; [6]Genetics, Genomics, and Bioinformatics Graduate Program, University at Buffalo and New York State Center of Excellence in Bioinformatics and Life Sciences, Buffalo, United States; [7]Department of Bioengineering, Stanford University, Stanford, United States; [8]Beijing Innovation Center for Genomics, Peking University, Beijing, China; [9]Biomedical Pioneering Innovation Center, Peking University, Beijing, China; [10]Prevail Therapeutics- a wholly-owned subsidiary of Eli Lilly and Company, New York, United States; [11]Department of Biochemistry and Molecular Biophysics, Vagelos College of Physicians and Surgeons, Columbia University Irving Medical Center, Columbia University, New York, United States

**\*For correspondence:**
sl682@cumc.columbia.edu

**Abstract** Olfactory receptor (OR) choice represents an example of genetically hardwired stochasticity, where every olfactory neuron expresses one out of ~2000 OR alleles in the mouse genome in a probabilistic, yet stereotypic fashion. Here, we propose that topographic restrictions in OR expression are established in neuronal progenitors by two opposing forces: polygenic transcription and genomic silencing, both of which are influenced by dorsoventral gradients of transcription factors NFIA, B, and X. Polygenic transcription of OR genes may define spatially constrained OR repertoires, among which one OR allele is selected for singular expression later in development. Heterochromatin assembly and genomic compartmentalization of OR alleles also vary across the axes of the olfactory epithelium and may preferentially eliminate ectopically expressed ORs with more dorsal expression destinations from this 'privileged' repertoire. Our experiments identify early transcription as a potential 'epigenetic' contributor to future developmental patterning and reveal how two spatially responsive probabilistic processes may act in concert to establish deterministic, precise, and reproducible territories of stochastic gene expression.

## eLife assessment

This is an **important** paper that revises the canonical model of how olfactory sensory neurons choose which odor receptor to express. The data presented in the paper are **convincing** and the model proposed is provocative and likely to enable important future work.

## Introduction

The development of multicellular organisms relies on gene expression programs that are precisely regulated in space and time. To transform probabilistic biochemical reactions, such as transcription and translation, into reproducible differentiation processes, plants, and animals convert individual cellular variability into predictable cell population averages. Yet, there are cases in biology where gene expression variability is desirable, as it generates diverse cellular identities that are difficult to obtain with deterministic gene regulation. For example, the production of antibodies via VDJ recombination, and evasion of immunological responses by antigenic variation represent biological systems that seek utmost randomness (*Ba et al., 2020*; *Borst, 2002*). Other biological functions, however, benefit from balancing absolute determinism with complete randomness, producing biased stochasticity. Genetically encoded biased stochasticity is often observed in the nervous system, where gene expression choices generated by neurons must integrate into functional and reproducible circuits (*Courgeon and Desplan, 2019*). In fly ommatidia, for example, biased randomness preserves a ratio of photoreceptor neuron identities across animals (*Johnston and Desplan, 2014*), whereas in mammals, random Protocadherin promoter choice (*Canzio and Maniatis, 2019*), was recently shown to obey spatial patterns in the mouse neocortex, assuring proper tiling between neighboring neurons (*Lv et al., 2022*).

Mammalian OR gene choice provides an extreme case of hardwired biased randomness (*Buck and Axel, 1991*). OR transcription starts in neuronal progenitors of the main olfactory epithelium (MOE), which transiently express 5–15 ORs out of >1000 OR genes distributed in genomic clusters across chromosomes (*Hanchate et al., 2015*; *Tan et al., 2015*; *Saraiva et al., 2015*). As these progenitor cells differentiate into post-mitotic olfactory sensory neurons (OSNs), they switch from polygenic to monogenic and monoallelic OR transcription (*Chess et al., 1994*). This transition is mediated by the assembly of a multi-chromosomal enhancer hub over the chosen OR allele (*Lomvardas et al., 2006*; *Markenscoff-Papadimitriou et al., 2014*; *Monahan et al., 2019*), followed by the stabilizing effects of an OR-elicited feedback signal (*Lyons et al., 2013*; *Dalton et al., 2013*; *Serizawa et al., 2005*; *Shykind et al., 2004*; *Lewcock and Reed, 2004*). During this developmental progression, heterochromatic silencing (*Magklara et al., 2011*) and genomic OR compartmentalization (*Clowney et al., 2012*; *Le Gros et al., 2016*) act together to assure that the non-chosen OR alleles will remain transcriptionally repressed for the life of the OSN. Interestingly, the position of the OSN across the dorsoventral (DV) axis of the MOE predisposes this singular transcriptional choice towards a group of 50–250 OR genes (*Tan and Xie, 2018*), providing reproducible topographic restrictions in OR expression. The anatomical segments of the MOE that express a specific collection of OR identities are known as 'zones,' with their total number varying from 4 to 9, depending on the analyses and criteria used to define them (*Ressler et al., 1993*; *Vassar et al., 1993*; *Miyamichi et al., 2005*; *Zapiec and Mombaerts, 2020*; *Ruiz Tejada Segura et al., 2022*). Although zonal restrictions in OR expression have a well-established influence on the wiring of the olfactory circuit (*Sullivan et al., 1994*; *Sullivan et al., 1995*), the mechanisms that bias this singular transcriptional choice towards specific OR identities remain unknown.

Here, we identified genetically encoded mechanisms that introduce topographic biases in OR gene regulation. We report that OSN progenitor cells from various MOE segments transcribe OR mixtures consisting of ORs with the corresponding or with more dorsal expression identities. Ectopic expression of dorsal identity ORs at the polygenic stage of OR transcription is rectified during differentiation by preferential genomic silencing that is skewed towards ORs with more dorsal expression identities than the identity of the OSN. Patterns of polygenic OR transcription and genomic OR silencing are influenced by gradients of transcription factors NFI A, B, and X (*Gronostajski, 2000*). Triple NFI (NFIA, B, and X) deletion both eliminates heterochromatic silencing and genomic compartmentalization from ORs with ventral identities, as well as extinguishes their transcription in olfactory progenitors. Furthermore, spatial transcriptomics revealed a dramatic expansion of dorsomedial identity OR expression towards the ventral MOE and reciprocal transcriptional reduction of ventral identity ORs in triple NFI

knockouts (cKOs), suggesting that patterns of genomic OR silencing and polygenic OR transcription influence OR gene choice. Indeed, transcriptional induction of an OR allele in OSN progenitors biases the choice towards this allele in mature OSNs (mOSNs) throughout the MOE. Strikingly, by modulating the levels of OR induction in progenitor cells we can restrict the expression of this OR allele to more dorsal OSNs, where heterochromatic silencing and genomic compartmentalization are less prevalent. Thus, our studies reveal that position-responsive OR transcription in OSN progenitors may act as an 'epigenetic' signal for future singular choice among the previously transcribed ORs. Moreover, our data suggest that polygenic OR transcription and heterochromatic silencing/genomic compartmentalization could act as opposing regulatory 'heostats' that determine in a spatially influenced fashion the exact OR repertoire that is available for stochastic singular choice in mature OSNs.

## Results

### OSN progenitors co-transcribe an increasing number of zonal OR identities toward the ventral MOE

The mouse MOE is divided into a limited series of stereotypic segments, based on the expression of OR genes, that exhibit bilateral symmetry between the two nasal cavities (*Figure 1A*). In whole-mount views, these segments present a DV segmentation pattern, with zone 1 being at the dorsal and zone 5 at the ventral end of the MOE. Intricate invaginations of the MOE occurring during embryonic development and early postnatal growth convolute this dorsoventral segmentation pattern, especially when viewing coronal sections of the MOE (*Figure 1A*). However, we will continue referring to the DV coordinates of each one of the five segments, or zones, as they correspond to their initial patterning during development.

Within each zone, mOSNs express a single OR allele among 50–250 OR genes with proper zonal identities. However, before the onset of singular OR expression, mitotically active OSN progenitors, the immediate neuronal precursor (INP) cells, co-express multiple lowly expressed ORs (*Hanchate et al., 2015*; *Tan et al., 2015*; *Saraiva et al., 2015*). To determine whether zonal restrictions are operational from this polygenic stage of OR transcription, we performed plate-based single-cell RNA-seq (scRNA-seq) analysis of FAC-sorted OSNs and OSN progenitors isolated from the MOE, which was micro-dissected into two parts: a more dorsal (zone 1–2) and a more ventral (zone 3–5) segment. With a median of over 130,000 unique transcripts per cell, this method readily detected low level OR gene transcription in OSN progenitors. Additionally, to enrich our plate-based scRNA-seq for cell populations of interest we used *Mash1-CreER*; Cre-inducible tdTomato reporter; *Ngn1-GFP* triple transgenic mice (*Figure 1—figure supplement 1A*). We injected P2 mice with tamoxifen, inducing permanent tdTomato expression, and then collected cells 48 hours later (*Figure 1—figure supplement 1A*). From each dissection we isolated four major cellular populations corresponding to four successive differentiation stages, as previously described (*Fletcher et al., 2017*; *Gadye et al., 2017*): GBCs (MOE stem cells), INPs (immediate neuronal precursors), iOSNs (immature OSNs), and mOSNs (*Figure 1—figure supplement 1A–B*).

Dimensionality reduction and clustering of single-cell RNA-seq data with Seurat (*Satija et al., 2015*) sorted cells into six populations. We determined each population's stage of OSN development using the expression of known MOE markers (*Figure 1B*, *Figure 1—figure supplement 1B–C*). We first detect OR mRNAs in INP3 cells (*Figure 1B*), which consistently transcribe multiple ORs. Next, we examined the zonal identity of ORs co-transcribed within individual dorsal or ventral INP3 cells. We created a separate category for the ~100 'class I' OR genes, which were grouped based on homology and resemblance to OR genes first identified in fish. Virtually all class I OR genes are expressed in zone 1, but their expression is likely regulated though a separate mechanism (*Hirota et al., 2007*; *Enomoto et al., 2019*), thus we chose to analyze them separately. Surprisingly, while dorsally positioned INPs transcribe almost exclusively dorsal identity ORs, ventrally positioned INPs transcribe complex mixtures consisting of ORs of every zonal identity (*Figure 1C–D*). Overall, ventral INPs transcribe a greater number of OR genes compared to dorsal INPs. Focusing on OR genes detected with at least three unique transcripts, we detect dorsal identity ORs in 43 ventral INP cells and ventral identity ORs in only 29 of them, while dorsal INP cells express predominantly dorsal ORs (*Figure 1C–D*). Moreover, as the ventral INPs differentiate to iOSNs, dorsal identity OR transcription is replaced by the 'correct' (zone-appropriate) ventral OR transcription, culminating in singular expression of an OR

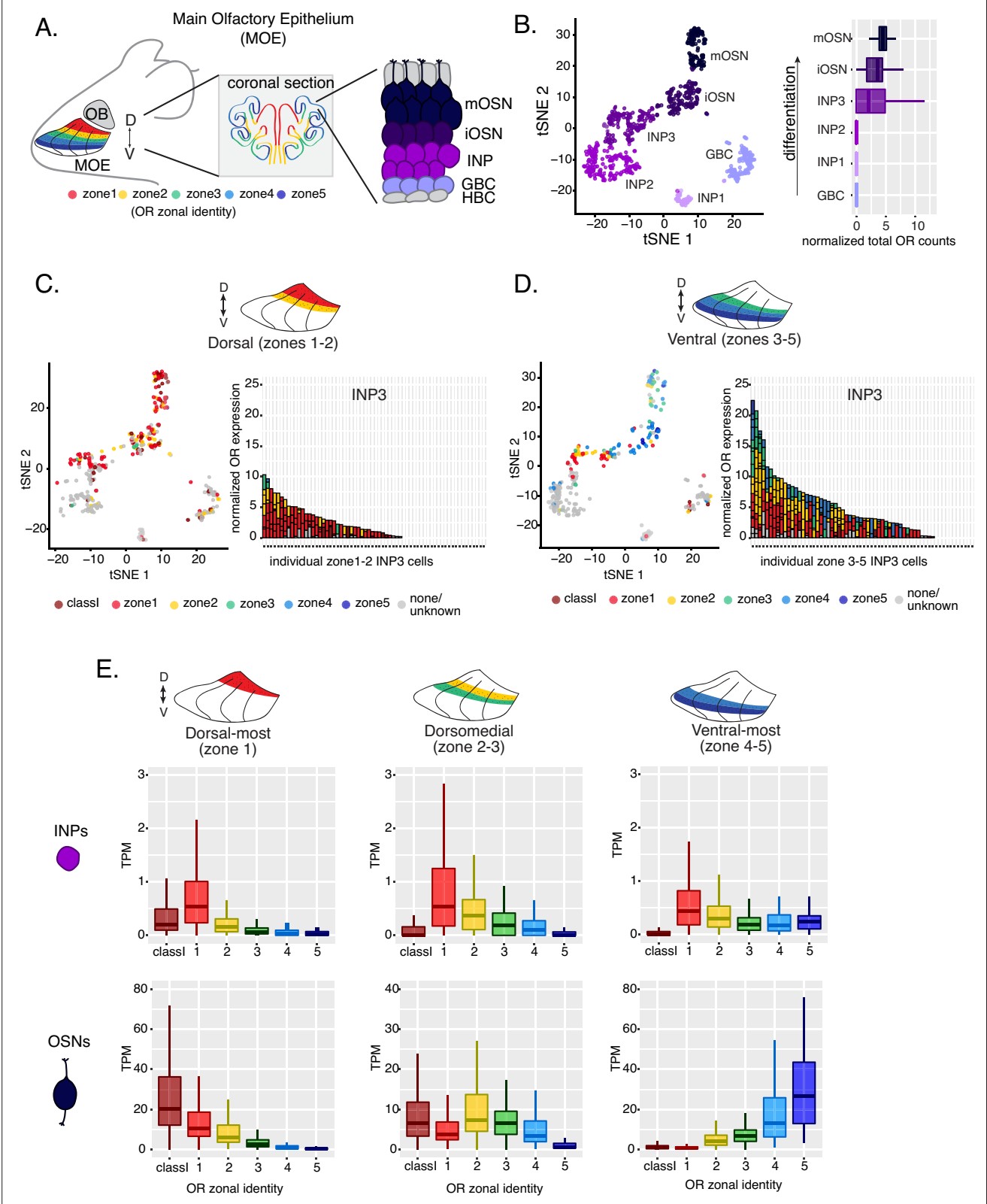

**Figure 1.** Polygenic transcription of olfactory receptor (OR) genes in olfactory progenitors follows a zonal expression pattern. (**A**) Schematic illustrating OR zones along the dorsoventral axis in whole mount views of the MOE (left) and coronal sections (middle). Zone 1 (red) is the dorsal-most zone and zone 5 (blue) is the ventral-most zone. Zoomed-in view of the MOE (right) shows cell populations at different stages of olfactory sensory neuron (OSN) differentiation organized in a pseudostratified fashion from the basal (least differentiated) to apical (most differentiated) layers: HBC, horizontal basal

*Figure 1 continued on next page*

*Figure 1 continued*

cell; GBC, globose basal cell; INP, immediate neuronal precursor; iOSN, immature olfactory sensory neuron; mOSN, mature olfactory sensory neuron. MOE: main olfactory epithelium, OB: olfactory bulb. (**B**) t-SNE (t-distributed Stochastic Neighborhood Embedding) dimensionality reduction used to visualize the clustering of single cells from FAC-sorted MOE cell populations with Seurat, based on expression of the most variable genes. Plot (left panel) shows the separation of single cells into six populations, to which we assigned cell identities based on the expression of known MOE markers (*Fletcher et al., 2017*) (See also *Figure 1—figure supplement 1*). Olfactory receptor expression is first detected in INP3 cells (right panel). (**C, D**, left panel) t-SNE plot (as in B) of cell populations isolated from either dorsal (zones 1–2) in (**C**) or ventral (zones 3–5) in (**D**) MOE microdissections. Cells are colored according to the zonal identity of the most highly expressed OR. (**C, D**, right panel) Plots depicting zonal identities of all the OR genes detected in individual INP3 cells from dorsal (**C**) or ventral (**D**) MOE. Y-axis shows OR expression in normalized counts of unique transcripts (UMIs) for different OR genes (separated by black lines). On the X-axis, each point is a different INP3 cell. ORs are colored according to their zonal identity. Note that while class I OR genes are expressed within zone 1 of the MOE, they may be regulated through a different mechanism and are thus displayed separately. Single-cell analysis shows data from two biological replicates. (**E**) Expression of OR genes of different zonal identities in olfactory progenitor INP cells (top) and mOSNs (bottom), determined with bulk RNA-seq, in cells isolated from dorsal-most (zone 1) (left), dorsomedial (zones 2–3) (middle), and ventral-most (zones 4–5) MOE microdissections (right). Note that INP and mOSN cells were FAC-sorted from the same exact dissection, thus the mOSN OR expression patterns confirm the accuracy of the dissection. RNA-seq was performed on INP and mOSNs from two or three biological replicates (from dorsal-most and ventral-most MOE segments or dorsomedial MOE segments, respectively).

The online version of this article includes the following figure supplement(s) for figure 1:

**Figure supplement 1.** Experimental strategy for isolating cells at different stages of olfactory sensory neuron development for single-cell RNA-seq.

allele with the correct zonal identity in mOSNs (*Figure 1D*). These observations were independently confirmed by bulk RNA-seq on FAC-sorted INP and mOSN cells extracted from trisected dorsal (zone 1), dorsomedial (zone 2–3), and ventral (zone 4–5) MOE, using the same labeling and FAC-sorting approach used for the single-cell experiments. This bulk analysis showed that in every case INPs co-transcribe ORs with the correct as well as more dorsal zonal identities, while further differentiation replaces dorsal ORs with ORs of the correct identity (*Figure 1E*). This finding immediately poses mechanistic questions about the process that shuts off dorsal ORs and enhances the transcription of the ORs expected to be expressed in each MOE segment.

## Heterochromatin eliminates ectopically expressed ORs along the dorsoventral MOE axis

We previously showed that OSN differentiation coincides with heterochromatin-mediated OR gene silencing (*Magklara et al., 2011*). If heterochromatinization contributes to singular OR choice by eliminating every non-chosen OR allele transcribed in INPs, then in any MOE segment silencing should be preferentially applied to ORs with the correct or more dorsal zonal identities. We performed native ChIP-seq in the MOE to determine the deposition of histone marks associated with heterochromatin including H3K9me3, a marker of constitutive heterochromatin, and H3K79me3, which we also found labeling heterochromatin on OR gene clusters (*Markenscoff-Papadimitriou et al., 2014*; *Monahan et al., 2017*). We predicted that dorsal-most identity (zone 1) ORs, which are expressed in INPs throughout the MOE, should have the highest levels of heterochromatin, whereas ventral-most identity (zone 5) ORs, which are transcribed only in ventral INPs, should have the lowest, with the rest of the OR repertoire having intermediate levels of heterochromatin marks. Indeed, visual inspection of ChIP-seq genomic tracks along OR gene clusters with mixed zonal constitution reveals the highest H3K9me3/H3K79me3 levels on the dorsal identity OR genes and the lowest on the ventral identity OR genes of the cluster (*Figure 2A*). Aggregate ChIP-seq analysis of all the OR genes grouped by their zonal identities corroborates the gradual reduction of H3K9me3 and H3K79me3 enrichment from dorsal to ventral ORs for the whole OR repertoire (*Figure 2B*). The only exception from this pattern is found at the dorsally expressed class I ORs, which rely on different regulatory mechanisms than the canonical class II ORs (*Hirota et al., 2007*; *Enomoto et al., 2019*; *Figure 2B*). Finally, using the FACS-based strategy described earlier, we confirmed that both heterochromatic marks are predominantly deposited during the INP to iOSN transition, simultaneously with the transition from polygenic to singular, zonally appropriate OR expression (*Figure 2C*, *Figure 2—figure supplement 1A*). Importantly, the descending pattern of heterochromatin enrichment from dorsal to ventral OR identities is preserved throughout differentiation.

We then asked if the patterns of heterochromatin deposition detected in mixed OSNs from the whole MOE are preserved in distinct zones. We performed ChIP-seq in mOSNs isolated from

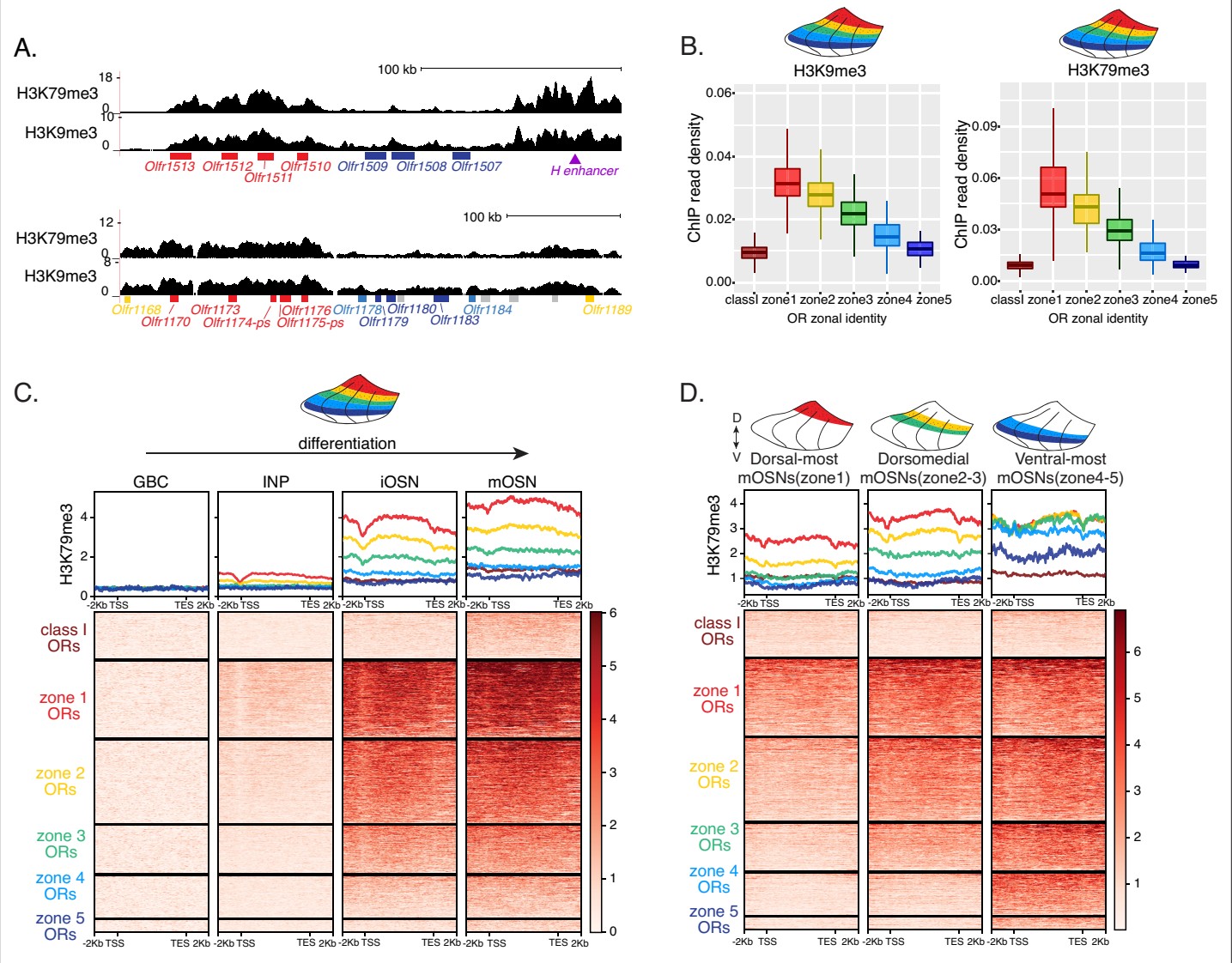

**Figure 2.** Heterochromatin deposition silences olfactory receptor (OR) genes from lower zones. (**A**) Signal tracks of H3K9me3 and H3K79me3 native ChIP-seq from the whole main olfactory epithelium (MOE) show heterochromatin deposition over two representative OR gene clusters. These clusters were selected because they harbor OR genes with both dorsal (zone 1) and ventral (zone 5) identities. OR genes are colored according to their zonal identity: zone 1 ORs in red, zone 2 ORs in yellow, zone 5 ORs in blue, and ORs with unknown zonal identity in gray. Purple triangle marks the 'H' OR gene enhancer that is present within that OR gene cluster. (**B**) H3K9me3 (left) and H3K79me3 (right) native ChIP-seq in the whole MOE. Box plots of read density over OR gene bodies, separated by their zonal identity, depict a pattern of deposition that is high on dorsal-most (zone 1) OR genes, progressively decreases with more ventral zonal OR identities, and is absent on class I ORs. (**C**) H3K79me3 native ChIP seq in globose basal cell (GBC), immediate neuronal precursor (INP), immature OSN (iOSN), and mature OSNs (mOSN) populations shows an onset of H3K79me3 deposition as cells transition from INPs to iOSNs. Each row of the heatmaps shows coverage over an OR gene body (separated into categories by their zonal identity). (See also *Figure 2—figure supplement 1A* for H3K9me3 heatmap). (**D**) H3K79me3 native ChIP-seq in mOSNs from zonally dissected MOE. Colored schematics above each heatmap depict the zone of dissection. (See also *Figure 2—figure supplement 1B* for H3K9me3 heatmap). (**A–D**) Pooled data from two biological replicates is shown for all ChIP experiments.

The online version of this article includes the following figure supplement(s) for figure 2:

**Figure supplement 1.** Progressive accumulation of heterochromatin on olfactory receptor (OR) genes in space and time.

dissected dorsal (zone 1), dorsomedial (zone 2–3), and ventral (zone 4–5) segments of the MOE. In each segment, OR genes with either the correct or more dorsal zonal identities had a higher level of heterochromatin compared to more ventral identity ORs (*Figure 2D*, *Figure 2—figure supplement 1B*). Intriguingly, this is the same zonal pattern observed for OR gene transcription in INP cells. Only

the OR gene identities able to be transcribed in the INP cells of a given zonal MOE segment acquire heterochromatin. Thus, most OR genes are heterochromatic in ventral OSNs; dorsal and dorsomedial identity OR genes are heterochromatic in dorsomedial OSNs; and only dorsal identity OR genes have some heterochromatin in dorsal OSNs (*Figure 2D*, *Figure 2—figure supplement 1B*). Although each zonal identity OR group is heterochromatic in the MOE segment where it is expressed by mOSNs, the level of H3K9me3/H3K79me3 enrichment is lower than in more ventral segments, where it is not chosen for stable expression. Thus, dorsal identity ORs have less heterochromatin in dorsal OSNs than in the rest of the MOE, and dorsomedial identity ORs have less heterochromatin in dorsomedial OSNs than ventral OSNs. Similarly, at the ventral end of the DV axis, ventral identity ORs have less heterochromatin than dorsal and dorsomedial identity ORs. Detection of heterochromatin on OR genes with the correct zonal identity is not counterintuitive, as only one OR allele from the ones co-transcribed will be eventually chosen, and the rest must be silenced. Thus, in a pure population of ventral mOSNs expressing *Olfr1507* (a ventral, zone 5 identity OR), the remaining non-chosen zone 5 identity OR genes are silenced with the same level of heterochromatin as OR genes with more dorsal zonal identities (*Figure 2—figure supplement 1C*). In other words, in every MOE segment, OR heterochromatinization is preserved only for the ORs that have the potential to be expressed and is not applied to more ventral ORs, which were not transcriptionally active in INPs. This is consistent with recent reports of heterochromatin marks being detected on trace amine-associated receptor (TAAR) genes only in TAAR-expressing OSNs and not the rest of the MOE (*Fei et al., 2021*).

## DV gradient of OR gene compartmentalization follows patterns of heterochromatin assembly

Heterochromatic OR genes converge into multi-chromosomal genomic aggregates of extreme chromatin compaction that contribute to the effective and stable silencing of non-chosen ORs (*Clowney et al., 2012*). We thus asked if the spatially determined pattern of OR heterochromatinization at the linear genome coincides with a similar pattern of 3D genomic compartmentalization. We performed in situ Hi-C in FAC-sorted mOSNs isolated from MOE segments along the DV axis and determined the frequency with which OR genes form long-range interactions in each segment. We saw a striking resemblance between the deposition of heterochromatic marks on OR genes and genomic compartmentalization (*Figure 3A–B*, *Figure 2—figure supplement 1D*). For example, inspection of the long range *cis* genomic interactions between 3 OR gene clusters on chromosome 2 shows that a cluster of predominately ventral identity OR genes is recruited to OR compartments only in ventral OSNs, where they are heterochromatic (*Figure 3A*). In contrast, the other two OR gene clusters, which are either enriched for dorsal ORs, or have mixed constitution, make strong genomic contacts with each other in all three MOE segments (*Figure 3A*). To expand this analysis to every OR, we measured the frequency of interchromosomal *trans* genomic interactions between OR genes with different DV identities. To do so, we annotated each OR gene cluster bin at 50 kb resolution according to the zonal identity of the residing ORs and plotted the average interchromosomal contacts between bins of different constitution, excluding class I OR genes, which formed very few contacts with other ORs. This analysis yielded the same conclusion: Interactions between dorsal OR genes is the default in every OSN, whereas compartmentalization for the remaining of the OR repertoire increases along the DV axis of the MOE (*Figure 3B*). Intriguingly, as with levels of heterochromatin, we detect the following recurrent pattern of OR compartmentalization: every OR has intermediate Hi-C contact frequencies with other OR genes in their segment of expression in the MOE, lower Hi-C contact frequencies in more dorsal MOE segments, and higher Hi-C contacts in more ventral segments.

The 'intermediate' levels of heterochromatin enrichment and Hi-C contacts observed on OR genes within their segment of expression in the MOE may reflect a less compact, transcription-compatible state of heterochromatin, or less frequent silencing of these OR genes compared to more dorsal identity ORs. To distinguish between the two scenarios, we explored OR silencing at the single-cell level using Dip-C, a single-cell Hi-C method (*Tan et al., 2018*; *Longzhi et al., 2020*; *Tan et al., 2019*). We performed Dip-C in 48 dorsal and 48 ventral mOSNs (*Figure 3C*). To retain allelic information, we used heterozygous mice from a cross between *Omp-IRES-GFP* (a mOSN reporter) and Castaneous (Cas) mice, and used Cas-specific SNPs to distinguish Cas from non-Cas alleles. Analyzing single-cell genomic contacts, we saw a greater enrichment of contacts between OR gene loci in ventral cells relative to dorsal cells, consistent with our bulk Hi-C data (*Figure 3D*). We then used the haplotype

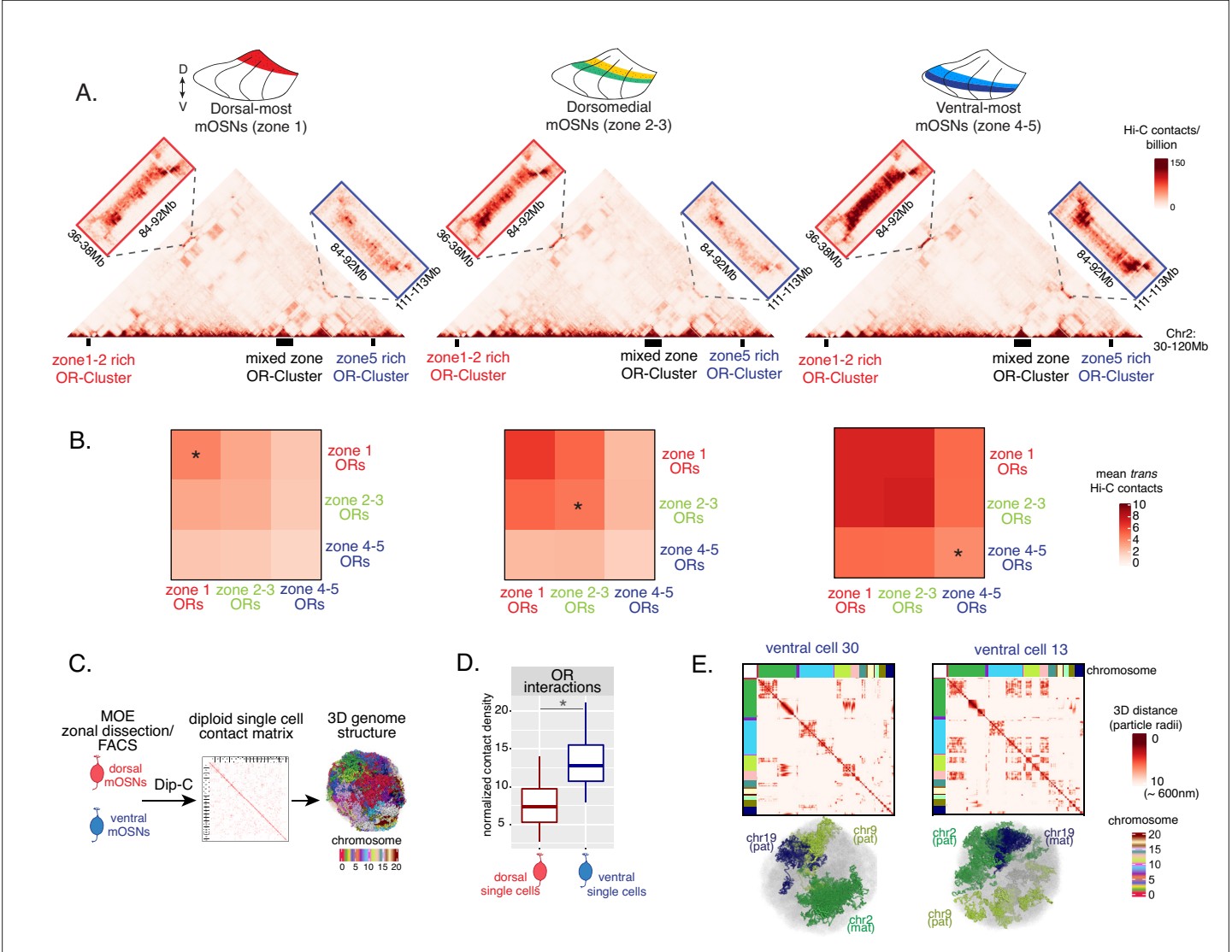

**Figure 3.** Zonal olfactory receptor (OR) compartmentalization permits OR genes from more ventral zones to be recruited into the OR compartment. (**A**) In situ Hi-C contact matrices of a 90 Mb region of chromosome 2 that contains three large OR gene clusters, depicted with boxes under the contact matrices. Hi-C libraries were prepared from mature OSNs (mOSNs) FAC-sorted from dorsal-most (zone 1) and ventral-most (zone 4–5) main olfactory epithelium (MOE) microdissections, as well as a pure population of Olfr17 (a zone 2 OR) expressing dorsomedial mOSNs. For each zonal contact matrix, magnified views show long-range *cis* Hi-C contacts between the large OR gene cluster in the middle that contains ORs of every zonal identity with the OR cluster on the left that contains mostly zone 1–2 identity ORs (red box) and the OR cluster on the right that contains mostly zone 4–5 identity ORs (blue box). *Cis* contacts between OR genes increase from dorsal to ventral mOSNs, but the zone 4–5 identity OR cluster associates with the other ORs only in the ventral-most OSNs (as seen when comparing Hi-C contacts in the blue boxes). (**B**) Heatmaps of average interchromosomal Hi-C contacts between OR genes annotated by their zonal identity at 50 Kb resolution show increased *trans* contacts in mOSNs from more ventral zones. OR genes have a similar, intermediate frequency of contacts in the mOSN population where they are expressed, marked with an asterisk. Class I OR genes (which are also expressed in zone 1) make few interchromosomal interactions in all zones (data not shown) and were thus excluded from this analysis. (**A–B**) Pooled data from two biological replicates is shown for all Hi-C analysis. (**C**) Dip-C in mOSNs from dorsal and ventral dissected MOE was used to generate haplotype-resolved single-cell contact matrices and 3D genome structures, as previously described (***Tan et al., 2019***). (**D**) Analysis of Dip-C contact densities of interchromosomal contacts between ORs genes confirms that ventral mOSNs have increased OR compartment interactions. Wilcoxon rank sum test: p-value = 9.164e-11. (**E**) Single-cell heatmaps of pairwise distances between OR genes generated from 3D genome structures in two ventral mOSNs show OR genes from different chromosomes intermingle in a different pattern in the two cells (top). For each cell, heatmaps are sorted by chromosome order and show all OR interactions within 10 particle radii (approximately ~600 nm). Representative 3D structures show the different positioning of three chromosomes (chr19, chr9, and chr2) in the two cells, resulting in a different pattern of OR cluster contacts (bottom). See also ***Figure 3—figure supplement 1*** for heatmaps of Dip-C distances in each of the 48 dorsal and ventral mOSNs.

The online version of this article includes the following figure supplement(s) for figure 3:

*Figure 3 continued on next page*

*Figure 3 continued*

**Figure supplement 1.** Olfactory receptor (OR) compartments are highly variable between single cells but show a consistent difference between cells of different zones.

resolved chromatin contacts to compute distances of all genomic loci at 20 kb resolution, from which we generated 3D genome structures for all cells (*Figure 3C*), as previously described (*Tan et al., 2019*). Analyzing distances between pairs of OR loci in the 3D genome structures we determined the size and complexity of OR compartments in each cell. We confirmed that OR compartments in ventral mOSNs are larger and contain more OR genes from more chromosomes than in dorsal mOSNs (*Figure 3—figure supplement 1C*). Importantly, in each cell, significantly fewer ventral identity OR genes were found in OR compartments compared to dorsal (zone 1) or dorsomedial (zone 2–3) identity OR genes (*Figure 3—figure supplement 1D*). From this result we can conclude that the 'stronger' Hi-C contacts among dorsal ORs observed in bulk, represent an increased number of dorsal ORs participating in OR compartments in each OSN. Thus, extrapolating Dip-C results to H3K9me3/H3K79me3 enrichment, we conclude that 'intermediate' silencing levels of each OR group in their own zone likely reflect less frequent silencing of these ORs than ORs with more dorsal zonal identities. In this note, OR compartmentalization is highly probabilistic, with each one of the 48 dorsal and ventral OSNs having unique maps of OR-OR genomic interactions (*Figure 3E*, *Figure 3—figure supplement 1A–B*). Thus, we propose that the balance between two probabilistic yet DV-responsive processes, early transcription and genomic silencing, may define the OR ensemble that is available for singular choice along the DV axis. To test this model, we sought to identify factors responsible for generating these remarkable patterns.

## NFI paralogues generate DV patterns in OR expression

We searched our RNA-seq data for transcription factors that have strong expression during the INP to iOSN transition that is graded across the DV axis of the MOE. NFI paralogues *Nfia*, *Nfib*, and *Nfix* have strong, DV-influenced expression in INPs that is preserved in iOSNs (*Figure 4A and B* and *Supplementary file 1*). Specifically, *Nfia* and *Nfib* are expressed higher in ventral INPs and iOSNs, and *Nfix* is higher in ventral mOSNs (*Figure 4B*). These three members of the nuclear factor I (NFI) family of transcription factors control a plethora of developmental and cell specification processes (*Gronosta-jski, 2000*; *Zenker et al., 2019*), and were previously implicated in OSN differentiation (*Baumeister et al., 1999*; *Behrens et al., 2000*). Thus, we decided to genetically explore their contribution in the establishment of dorsoventral patterns of OR expression.

To interrogate the potential role of NFIA, B, and X in zonal OR expression we deleted all three genes simultaneously using the *Krt5-CreER* driver, which is expressed in the quiescent stem cells of the MOE (HBCs). We crossed *Krt5-CreER*; Cre-inducible tdTomato reporter mice to *Nfia*, *Nfib*, *Nfix* triple fl/fl mice (*Clark et al., 2019*), and induced recombination with tamoxifen. To force the quiescent HBCs to differentiate into OSNs, we ablated the MOE with methimazole and allowed 40 days for a complete restoration by the marked progeny of the NFI triple conditional knockout (cKO) or control HBCs (*Figure 4—figure supplement 1A–B*), as previously described (*Monahan et al., 2019*). RNA-seq analysis of the FAC-sorted cKO OSNs from the whole MOE revealed significant transcriptional reduction of ventral OR identities and reciprocal increase of dorsomedial ORs (*Figure 4C*). In contrast, triple NFI deletion only in mOSNs, using the *Omp-IRES-Cre* driver, has no measurable effects on OR expression (*Figure 4D*). To determine whether the reduced transcription of ventral ORs reflects a developmental defect of ventral OSN differentiation, versus a *bona fide* dorsalization of ventral OSNs, we performed RNA-seq in cKO OSNs isolated specifically from ventral MOE microdissections. This experiment revealed ectopic expression of OR genes with dorsomedial zonal identities in place of the OR genes with proper ventral ones (*Figure 4E*), a result confirmed by immunofluorescence (IF) experiments (*Figure 4—figure supplement 1D–E*). This transcriptional transformation of ventral OSNs satisfies the original criteria of homeosis (*Bateson, 1894*), since the overall mOSN identity is not altered by the triple NFI deletion: Only 13 out of ~200 OSN-specific genes are significantly different with at least a twofold change between control and cKO OSNs, and 113/207 non-OR ventral markers are still highly expressed in the ventral-most zones, acting as independent fiducial markers for our zonal dissection (*Figure 4—figure supplement 1F*). Interestingly, deletion of just one allele of each *Nfia*, *Nfib*, and *Nfix* had an intermediate effect, with expression of ventral-most (zone 5) ORs replaced

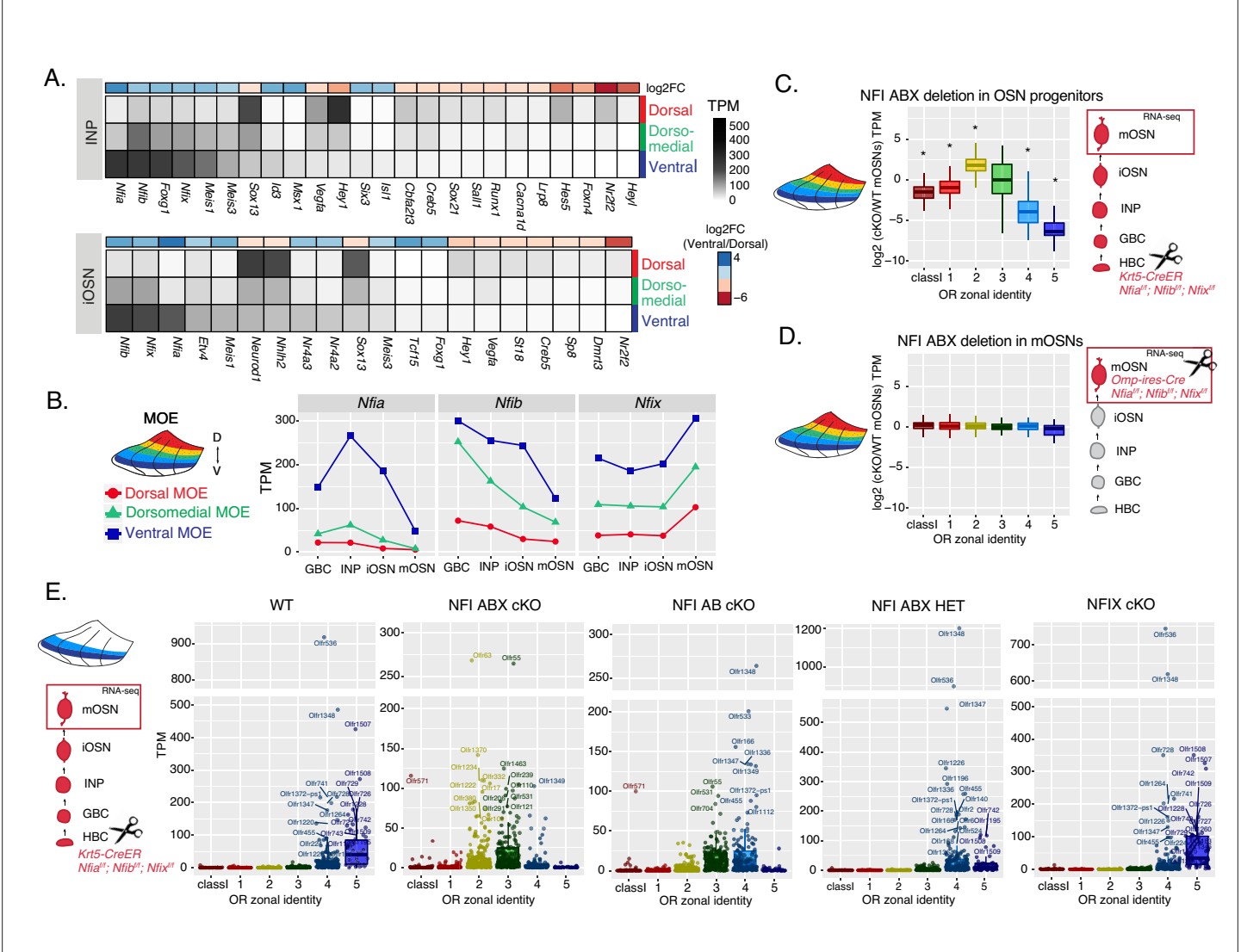

**Figure 4.** NFI paralogue gradients regulate zonal olfactory receptor (OR) expression. (**A**) Heatmaps showing differentially expressed transcription factors in the immediate neuronal precursor (INP) and immature olfactory sensory neuron (iOSN) cells isolated from the either dorsal, dorsomedial, or ventral olfactory epithelium. The shown transcription factors are significantly differentially expressed between dorsal and ventral cells with an adjusted p-value of <0.05, at least a threefold change in expression, and an expression level of at least 15 TPM (transcripts per million). Adjusted p-values use the Benjamini-Hochberg method to control for multiple hypothesis testing. A broader list of zonal transcription factors is included in *Supplementary file 1*. The heatmaps are sorted based on expression in ventral cells and the color bar above each heatmap shows the log2 fold change in ventral cells relative to dorsal cells. Three biological replicates were analyzed for INP and iOSNs from dorsal-most and ventral-most main olfactory epithelium (MOE), and two biological replicates were analyzed for INP and iOSNs from dorsomedial MOE. (**B**) Expression levels of *Nfia, Nfib,* and *Nfix* at four stages of olfactory sensory neuron (OSN) development in dorsal cells (red), dorsomedial cells (green) and ventral cells (blue). (**C, D**) Comparison of OR gene expression in NFI ABX triple knockout (*Nfia, Nfib,* and *Nfix* deletion) and control cells from the whole MOE. NFI transcription factors are deleted either in olfactory progenitors (**C**) using the *Krt5-CreER* driver or in mOSNs (**D**) using the *Omp-IRES-Cre* driver (as illustrated in *Figure 4—figure supplement 1*). At the right of each panel, scissors indicate the differentiation stage of *Nfia, Nfib,* and *Nfix* deletion, and a red box marks the cell type that was FAC-sorted for RNA-seq analysis. Two biological replicates were compared for NFI ABX triple knockout in olfactory progenitors and controls (**C**), and three biological replicates were compared for NFI ABX triple knockout in mOSNs and controls (**D**). Wilcoxon rank sum test: *p-value <0.01 [Benjamini-Hochberg FDR = 0.05]. (**E**) OR expression in NFI ABX triple knockout, NFI AB double knockout, NFIX knockout, NFI ABX triple heterozygous and control mOSNs from ventrally dissected MOE. Knockout was induced in progenitors with the *Krt5-CreER* driver. Plots show a different pattern of OR gene transcription in the different genotypes. Quantification of differentially expressed ORs for the three knockout genotypes is shown in *Figure 4—figure supplement 1*. Three biological replicates were compared for NFI ABX triple knockout mOSNs, two replicates for NFI AB double knockout, NFIX knockout, and NFI ABX triple heterozygous mOSNs, and four replicates for control mOSNs.

The online version of this article includes the following figure supplement(s) for figure 4:

**Figure supplement 1.** NFI ABX deletion in zone 5 olfactory epithelium results in a shift in the olfactory receptor (OR) repertoire.

with zone 4 ORs in ventral mOSNs (*Figure 4E*). In fact, the severity of this dorsomedial transformation depends on the total number of NFI alleles deleted, with the triple *Nfia*, *Nfib*, *Nfix* deletion (NFI ABX cKO) mOSNs expressing predominantly zone 2 and 3 ORs, double *Nfia*, *Nfib* deletion (NFI AB cKO) mOSNs expressing zone 3 and 4 ORs, and single *Nfix* deletion (NFIX cKO) mOSNs having almost wild type expression patterns of zone 4 and 5 ORs (*Figure 4E*, *Figure 4—figure supplement 1C*).

## Spatial transcriptomics reveals widespread homogenization and dorsalization of the MOE upon triple NFI deletion

To obtain a complete and unbiased understanding of the consequences of triple NFI deletion on patterns of OR expression, we deployed a spatial transcriptomic approach. Since our goal was to decipher zonal patterns of OR expression across the dorsoventral MOE axis without requirements for single-cell resolution, we opted for the Visium Spatial Gene Expression workflow (10 X Genomics) (*Ståhl et al., 2016*). This workflow is ideal for the interrogation of spatial OR expression in mOSNs, as OR mRNAs are highly abundant and readily detectable in most spatial spots that contain OSN mRNAs. For increased stringency, we only included spatial spots that include more than two OR genes and three OR transcripts. We analyzed 2 MOE sections each from an NFI ABX triple cKO and age-matched control mouse (*Figure 5A*). Expression data on OR genes were normalized and integrated across replicates (see methods). We performed PCA analysis, by which spatial spots were arranged in five clusters in control and cKO MOEs (*Figure 5B*). Interestingly, while dimensionality reduction and unbiased clustering generated OR clusters that correspond to zonal patterns of OR expression (i.e. each cluster contains OR genes with one zonal identity) in control MOEs, only dorsal-most zone 1/ class I OR genes followed this correlation in cKO MOEs (*Figure 5B*). The other 4 clusters in the cKO homogenously express dorsomedial zone 2–4 ORs, with expanded expression of zone 2 ORs in every cluster and complete loss of ventral-most zone 5 OR expression. Thus, conditional triple NFI deletion causes loss of spatial patterning for zone 2–4 OR genes and loss of expression for zone 5 OR genes, without influencing the expression of zone 1 ORs.

To depict the effects of triple NFI deletion on spatial patterns of OR expression, we plotted the average OR expression per spatial spot of the top 20 most highly expressed OR genes with dorsal-most (zone 1), dorsomedial (zone 2), and ventral-most (zone 5) identities. We then overlaid the corresponding values against the histological images of the control (wt) and NFI ABX cKO MOEs (*Figure 5C*). As observed in the clustering and heatmap analysis, dorsal-most zone 1 OR expression is confined to the same anatomical region for both samples. However, dorsomedial zone 2 OR expression in the cKO MOE extends beyond its defined anatomical region from the control MOE, and spreads to the ventral-most zones (*Figure 5C*). This expansion is also observed in the expression of individual zone 2 OR genes (*Figure 5—figure supplement 1A*). In contrast, the top 20 zone 5 OR genes, while highly expressed in control MOEs, are almost undetectable in NFI cKO MOEs (*Figure 5C*), consistent with our RNA-seq analysis. Expression of *Olfr1507*, the most highly expressed zone 5 OR, is undetectable in the cKO spatial spots (*Figure 5—figure supplement 1B*), in agreement with our IF data. Finally, to obtain a more general understanding of the spatial transformations in OR expression patterning upon triple NFI deletion, we assigned a zonal identity to each spatial spot using the maximum normalized expression of all the OR genes detected within a spot (see methods). Unlike control MOEs, where spot assignment reproduces zonal anatomical positions, most spatial spots in the cKO MOEs, excluding the unchanged zone 1, are assigned a zone 2 identity. Even the few spots assigned a zone 3 identity are shifted towards more ventral positions within the MOE relative to control, in a striking dorsalization and homogenization of the MOE (*Figure 5D*).

## NFI gradients control patterns of OR heterochromatinization and polygenic OR transcription

We searched for a mechanistic explanation for the homeotic transformation of ventral OSNs in triple NFI cKO mice. Our experiments so far have identified three spatially responsive processes that may contribute to the dorsoventral patterning of OR gene choice: polygenic OR transcription in INPs, OR heterochromatinization, and genomic compartmentalization during the INP to iOSN transition. Thus, we explored the effects of triple NFI deletion in all three processes. First, we investigated the effects of NFI deletion on OR heterochromatinization with ChIP-seq in triple NFI cKO OSNs from the ventral-most MOE segments. ChIP-seq revealed an almost complete loss of heterochromatin from ventral

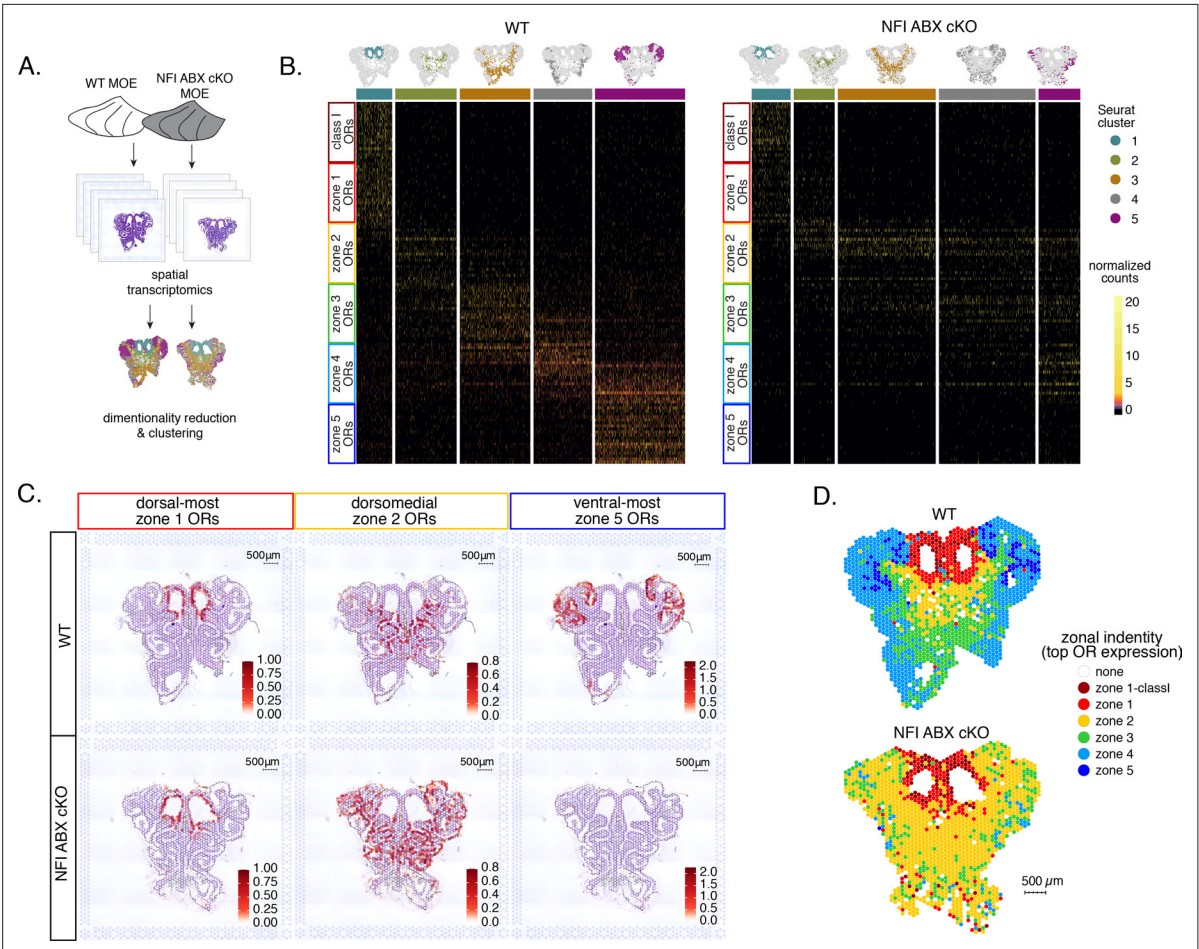

**Figure 5.** Spatial transcriptomics shows dorsalization and homogenization of the main olfactory epithelium (MOE) upon NFI A, B, and X deletion. (**A**) Schematic depicting our analysis pipeline: Spatial transcriptomics was performed on sections of wt control and NFI ABX conditional knockout (cKO) MOE. Dimensionality reduction was performed, and spatial spots were clustered based on normalized expression of olfactory receptor (OR) genes. (**B**) Heatmaps showing scaled, normalized expression levels of the top 20 highest expressed OR genes per zone in the control dataset. Unbiased neighborhood analysis and clustering grouped spatial spots into five clusters for both control and cKO MOE (depicted in distinct colors on the top of the heatmaps). Clustering of spatial spots in the control sample reproduces anatomical zones, as spots within each cluster express OR genes with the corresponding zonal identity (left heatmap). The same clusters were generated for NFI cKO sample (right heatmap). Although cluster 1 expresses exclusively zone 1 ORs, like in control MOEs, clusters 2–5 exhibit homogenous OR expression, with ventral expansion of zone 2/3 ORs, and reduced representation of zone 4/5 ORs. Heatmaps show deeply sequenced data from four sections from one mouse for wt and cKO sample. More shallowly sequenced data from four sections each from two biological replicates for wt and cKO sample showed similar results and are not shown. (**C**) Average normalized per-spot expression of the 20 highest expressed OR genes from zone 1, zone 2, and zone 5 is overlaid against H&E histological image of control (top) and NFI cKO (bottom) MOE sections. Expression of zone 1 OR genes is confined to the same anatomical region for both control and NFI cKO sections. Zone 2 OR gene expression is spread to more ventral regions in the NFI cKO compared to control sections, while expression of zone 5 OR genes is almost completely absent in the NFI cKO sample. (**D**) Spatial spots are colored according to their zonal assignment, which was determined based on the highest summed normalized expression of OR genes per zonal identity within that spot. Zonal spot assignment of the control sample visually reproduces known anatomical zones. In the NFI cKO sample, spots in the dorsal region have the highest expression of class I and zone 1 OR genes, similar to the control sample. However, in the rest of the NFI cKO MOE, most spots have a zone 2 OR identity. Spots assigned the identity 'none' did not contain any OR transcripts and were excluded from cluster analysis.

The online version of this article includes the following figure supplement(s) for figure 5:

**Figure supplement 1.** Expression of zone 2 olfactory receptors (ORs) genes spreads ventrally in NFI ABX knockout main olfactory epithelium (MOE).

ORs as well as a reduction on zone 3 dorsomedial ORs in NFI cKO ventral OSNs and INPs (***Figure 6A***, ***Figure 7—figure supplement 1F–G***). Similarly, in situ Hi-C in control and triple NFI cKO OSNs from ventral MOE segments revealed a strong reduction in the long-range *cis* and *trans* genomic contacts made by ventral ORs (***Figure 6B***). In contrast, dorsal ORs did not exhibit strong changes in ChIP-seq and Hi-C contacts (***Figure 6A and B***). Strikingly, in both processes, heterochromatin assembly and

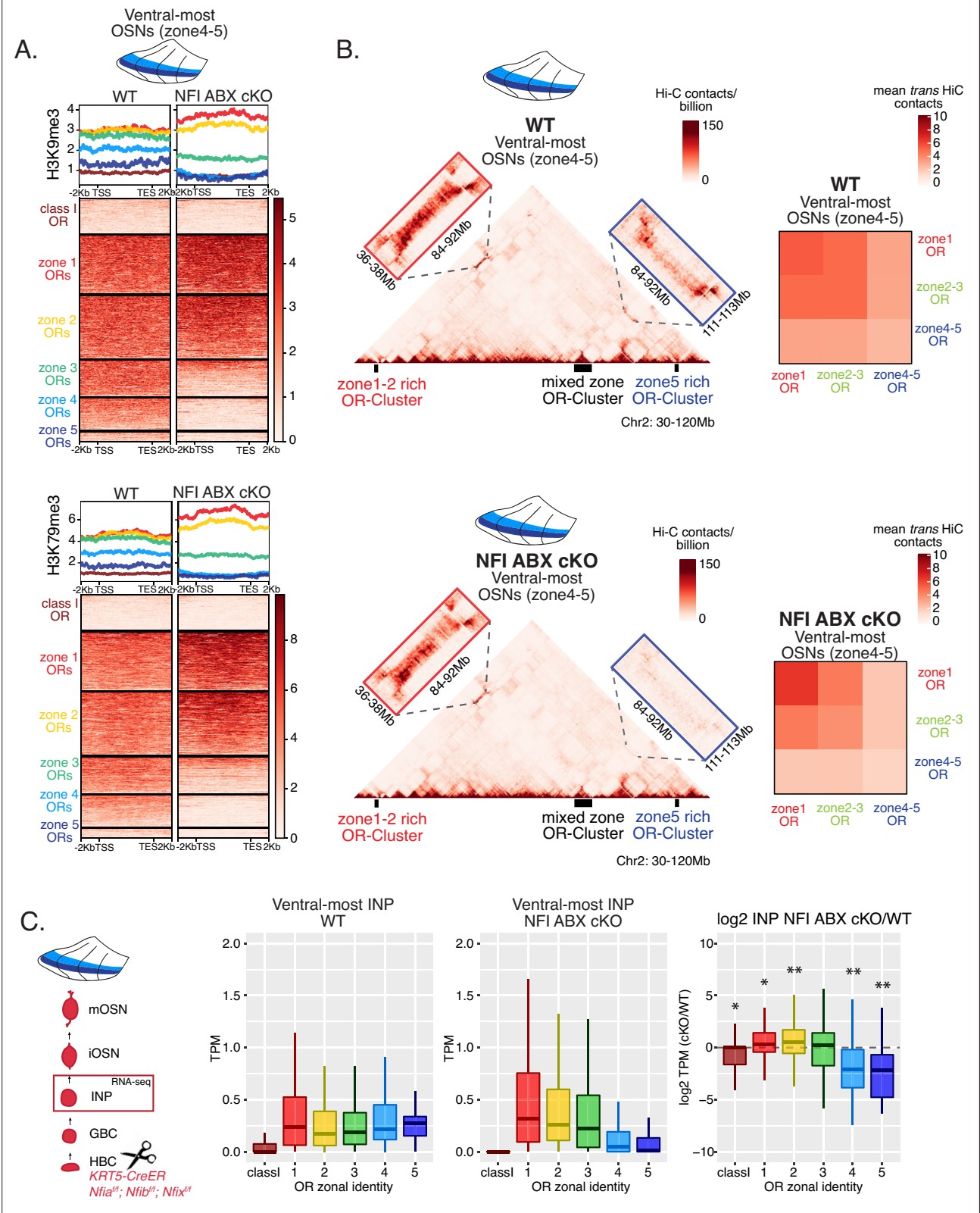

**Figure 6.** NFI A, B, and X regulate chromatin state and olfactory receptor (OR) compartment formation . (**A**) Native ChIP-seq for H3K9me3 (top) and H3K79me3 (bottom) in NFI ABX knockout mature OSNs (mOSNs) from ventral main olfactory epithelium (MOE). Heatmaps show ChIP signal over OR gene bodies, scaled to 6 kb with 2 kb flanking on either side. There is a decrease of both histone marks on zone 3–5 identity OR genes in NFI ABX knockout compared to control. Triple NFI deletion was induced with the Krt5-CreER driver (before OSN differentiation). (**B**) Hi-C in NFI ABX knockout

*Figure 6 continued on next page*

*Figure 6 continued*

and control mOSNs from ventral MOE. Left: In situ Hi-C contact matrices of a 90 Mb region of chromosome 2 from control (top) and NFI ABX triple knockout (bottom) ventral mOSNs, as described in *Figure 3A*. The contact matrix shows long-range *cis* interactions between three large OR gene clusters: one enriched for dorsal, zone 1–2, identity ORs (left), one containing ORs of every zonal identity (middle), and one enriched for ventral, zone 4–5, identity ORs. Note that long-range *cis* contacts between the zone 4–5 identity enriched cluster and the mixed identity cluster dissipate in the triple NFI cKO (bottom, blue box), whereas the contacts of the mixed identity cluster with the zone 1–2 identity enriched cluster are preserved (bottom, red box). Right: Heatmaps of average interchromosomal Hi-C contacts between OR genes annotated by their zonal identity (as described in *Figure 3B*) in control (top) and triple NFI cKO (bottom) mOSNs from ventral MOE. *Trans* contacts between zone 4–5 ORs dissipate, whereas *trans* contacts between zone 2–3 ORs reach intermediate levels typically detected between OR genes with the 'correct' zonal identity for a given MOE segment (see *Figure 3*). (**A–B**) Pooled data from two biological replicates is shown. (**C**) OR expression by zonal identity in immediate neuronal precursor (INP) cells isolated from ventral NFI ABX knockout and control MOE. Triple NFI deletion was induced with the Krt5-CreER driver (before OSN differentiation) and NFI ABX INP cells were isolated as described in *Figure 4—figure supplement 1*. Log2 fold change of OR expression in NFI ABX vs control INP cells shows a significant decrease in expression of zone 4–5 ORs (right). Wilcoxon rank sum test: *p-value <0.05, **p-value <0.001 [Benjamini-Hochberg FDR = 0.05]. Two biological replicates of NFI ABX cKO and control were analyzed.

The online version of this article includes the following figure supplement(s) for figure 6:

**Figure supplement 1.** Ventral NFI ABX cKO cells closely resemble dorsomedial cells in chromatin state and compartment formation.

genomic compartmentalization, the patterns observed in ventral OSNs upon NFI deletion are similar to those observed in dorsomedial OSNs from the control MOEs (*Figure 6—figure supplement 1A–B*).

Finally, we explored the effects of triple NFI deletion on the polygenic transcription of ORs in INP cells. We used a FACS-based strategy to isolate INPs from the ventral MOE followed by bulk RNA-seq as described earlier (*Figure 4—figure supplement 1A*). Again, as with the results from ChIP-seq and Hi-C experiments, we detect a conversion toward the signatures observed in dorsomedial INPs, i.e., detection of only dorsal and dorsomedial ORs and depletion of ventral OR identities from the INP transcriptome (*Figure 6C*). Thus, our data reveal an unexpected correlation between OR transcription in INP cells, and two diametrically opposing gene expression outcomes in OSNs: silencing for the majority of the co-transcribed OR alleles and singular choice for one of them. We devised a genetic strategy that would test the hypothesis that polygenic OR transcription is a pre-requisite for singular OR choice.

## Early OR transcription promotes OR gene choice in mOSNs

We manipulated OR transcription using a genetically modified *Olfr17* allele with a tetO promoter inserted immediately downstream of its transcription start site (*Fleischmann et al., 2013*). This allele enables strong transcriptional activation of *Olfr17* from the endogenous genomic locus under the control of tTA (*Figure 7A*, *Figure 7—figure supplement 1A*). *Olfr17* expression is monitored by an IRES-GFP reporter inserted immediately downstream of the *Olfr17* translational stop codon (*Figure 7A*). To induce transcription of this '*tetO-Olfr17*' OR allele in INPs and iOSNs, we used *Gng8-tTA* transgenic mice. *Gng8* is expressed in INPs and iOSNs, and completely shuts off in mOSNs (*Figure 7—figure supplement 1D*). Consistent with the expression properties of Gng8 and previous reports (*Nguyen et al., 2007*; *Nguyen et al., 2010*), we only detect GFP in the basal MOE layers of *Gng8-tTA*; *tetO-GFP* mice (*Figure 7B*), which are enriched for INP and iOSN cells. However, when we cross the same *Gng8-tTA* driver to *tetO-Olfr17* mice, we detect widespread GFP signal in apical MOE layers, which contain predominantly mOSNs (*Figure 7B*). Since there is no tTA expression in mOSNs, we reasoned that the INP/iOSN-induced tetO-Olfr17 allele is chosen for expression by the endogenous transcriptional machinery responsible for singular OR choice. Indeed, Hi-C experiments of these OSNs revealed that Greek Islands, the intergenic OR enhancers that converge over the chosen OR allele (*Lomvardas et al., 2006*; *Markenscoff-Papadimitriou et al., 2014*; *Monahan et al., 2019*), are recruited specifically to the *tetO-Olfr17* allele (*Figure 7C*), explaining the sustained expression of this OR in mOSNs. The hallmark of OR choice is the singular and stable expression of the chosen allele. Consistent with this, cells expressing the *tetO-Olfr17* allele do not express any other OR genes (*Figure 7—figure supplement 1E*). Furthermore, treating *tetO-Olfr17*; *Gng8-tTA* mice with high doxycycline (200 mg/kg in food) for 35 days fails to extinguish *tetO-Olfr17* expression in mOSNs (*Figure 7—figure supplement 1B–C*). Together these findings support the notion that transcriptional induction of *Olfr17* in INPs/iOSNs signals for the preferential choice of this OR in mOSNs.

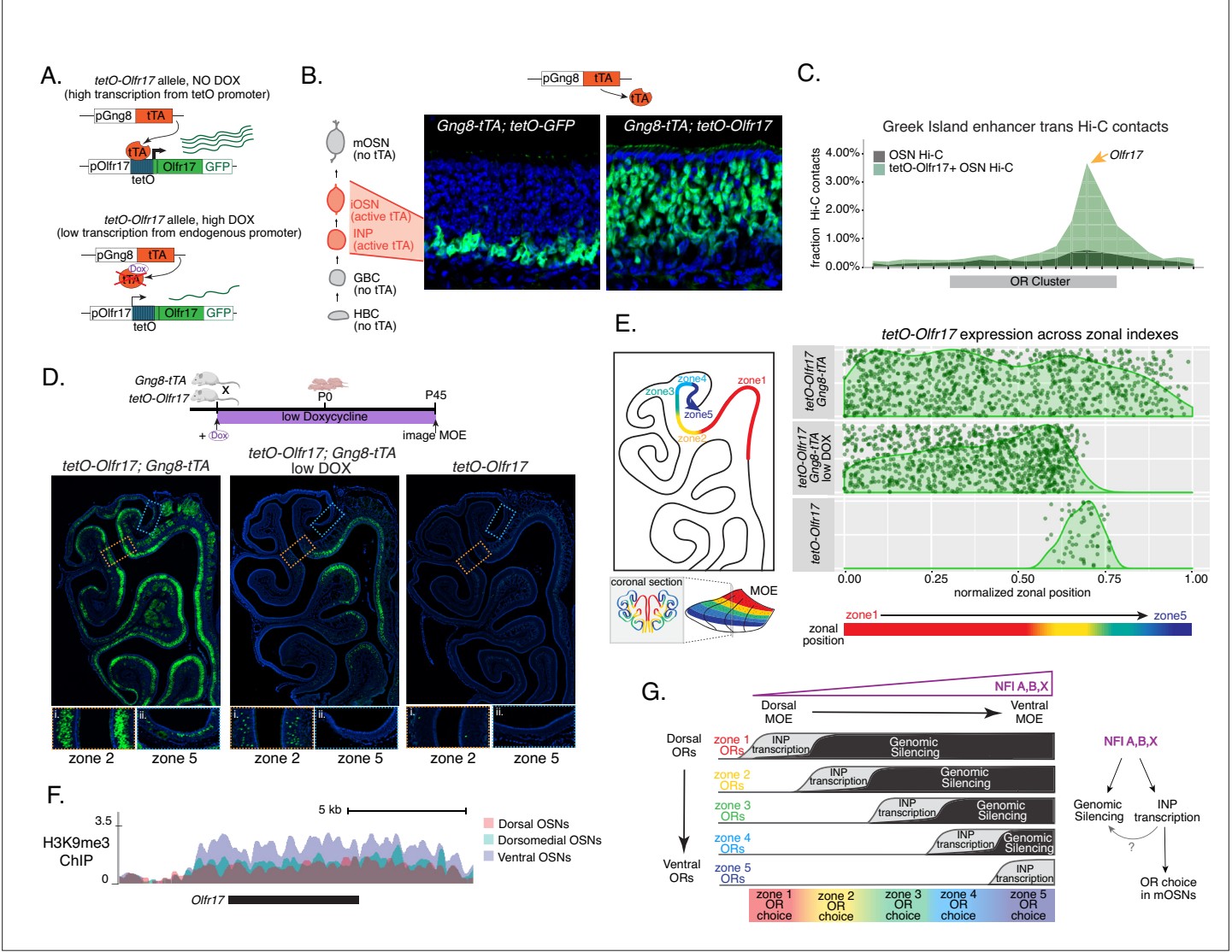

**Figure 7.** Genetic induction of olfactory receptor (OR) transcription in olfactory progenitors determines OR choice in mature OSNs (mOSNs). (**A**) Genetic strategy for transcriptional induction of OR *Olfr17* (a zone 2 identity OR) from its endogenous genomic locus. A genetically modified '*tetO-Olfr17*' allele contains a tetO promoter immediately downstream of the endogenous *Olfr17* promoter and an IRES GFP reporter after the coding sequence (*Fleischmann et al., 2013*). In the presence of tTA a high level of *tetO-Olfr17* is induced from the tetO promoter (top), while in the presence of a high amount of doxycycline (DOX) tTA is inhibited and transcription is regulated by the endogenous promoter. See also *Figure 7—figure supplement 1A* for information on the genomic locus of this *Olfr17* allele. (**B**) tTA driven by the *Gng8* promoter is expressed in immediate neuronal precursor (INP) and immature OSN (iOSN) cells in the main olfactory epithelium (MOE) (*Tirindelli and Ryba, 1996*). When *Gng8-tTA* drives the expression of a *tetO-GFP* allele, transcription is detected only in progenitor cells located on the basal side of the MOE, where the tTA is expressed (left) (*Nguyen et al., 2010*). In contrast, when *Gng8-tTA* drives the expression of *tetO-Olfr17*, expression persists in mature OSNs where tTA is no longer present (right). See also *Figure 7—figure supplement 1B–C* for the sustained and widespread expression of the *tetO-Olfr17* allele after 35 days of high DOX treatment and *Figure 7—figure supplement 1D* for *Gng8* expression during OSN differentiation. (**C**) In situ Hi-C in *tetO-Olfr17* expressing cells shows enriched contacts with interchromosomal olfactory receptor ('Greek Island') enhancers over the *Olfr17* locus, suggesting tetO-Olfr17 + OSNs are using endogenous mechanisms to sustain *Olfr17* expression after Gng8-tTA is no longer present. (**D**) *tetO-Olfr17* expression in coronal sections of the MOE determined by GFP fluorescence. In the absence of tTA *tetO-Olfr17* expression occurs only in zone 2 of the MOE (right); with high tTA induction in progenitor cells *tetO-Olfr17* expression occurs throughout all zones of the MOE (left); and with low tTA induction in progenitor cells, due to the addition of a low amount of doxycycline, *tetO-Olfr17* expression occurs in zone 2 and spreads dorsally to zone 1 (middle) only. Magnified views show *tetO-Olfr17* expression in its native zone 2 (**i**) and ectopic expression in the most ventral zone 5 (**ii**). Mice on low doxycycline (DOX) treatment were provided doxycycline at 1 ug/ml in water throughout gestation and postnatal life. (**E**) Quantification of *tetO-Olfr17* expression (determined by GFP fluorescence in immunofluorescence images) relative to a normalized zonal position (illustrated on the left) in coronal sections of the MOE from *tetO-Olfr17* without tTA driver (bottom), *tetO-Olfr17* with *Gng8-tTA* driver (top), and *tetO-Olfr17* with *Gng8-tTA* driver on low DOX (middle). Six sections from two replicates were analyzed for *tetO-Olfr17* with *Gng8-tTA*; 9 sections from two replicates were analyzed from *tetO-Olfr17* with *Gng8-tTA* and low DOX; 29 sections

*Figure 7 continued on next page*

*Figure 7 continued*

from two replicates were analyzed for *tetO-Olfr17* without tTA. The plot displays a maximum of 1000 cells randomly selected for each condition. (**F**) H3K9me3 native ChIP signal over the *Olfr17* locus in mOSNs from dorsal (red), dorsomedial (green), and ventral (blue) MOE shows a higher level of heterochromatin in ventral MOE. (**G**) Model of OR choice in each zone of the MOE, regulated by the interplay of low-level polygenic OR transcription in INP cells, which defines the OR repertoire that can be chosen in each zone, and heterochromatic silencing, which prevents ectopic expression of more dorsal ORs. Both polygenic OR transcription in INP cells and heterochromatin deposition are influenced by NFI A, B, and X transcription factors, expressed in a dorsal-low ventral-high gradient across the MOE.

The online version of this article includes the following figure supplement(s) for figure 7:

**Figure supplement 1.** Hijacking olfactory receptor (OR) gene choice by inducing OR transcription at the stage of polygenic OR transcription.

Intriguingly, transient induction of *Olfr17* transcription promotes preferential choice of this OR throughout the MOE, rather than only in zone 2, where *Olfr17* is normally chosen for expression (**Figure 7D**). In fact, the vast majority of mOSNs from zones 1–4 are GFP+, and only in the ventral-most zone 5 we detect a more sporadic pattern of ectopic *Olfr17* choice (**Figure 7D and E**). We hypothesized that reduced frequency of ectopic *Olfr17* expression in the most ventral segment reflects the fact that heterochromatin levels and genomic compartmentalization of this dorsomedial OR allele is highest at this MOE segment (**Figure 7F**). This immediately suggests that the balance between transcriptional activation and heterochromatic silencing during INP to iOSN transition determines whether an OR can be chosen for singular expression. If this hypothesis is correct, then reducing *Olfr17* transcription in INP/iOSN cells should preferentially prohibit ectopic *Olfr17* expression in ventral MOE segments, where heterochromatic silencing is stronger. To test this, we pharmacologically manipulated tTA activity using a low level of doxycycline (1 ug/ml in water) administered throughout gestation and postnatal life of the mouse (**Figure 7D**), which reduces but does not eliminate tTA-driven transcription. Remarkably, mice that were subjected to this doxycycline regimen continue to frequently express *Olfr17* in dorsal mOSNs (zones 1–2), but not in mOSNs from more ventral MOE segments (zones 3–5) (**Figure 7D and E**), where heterochromatin levels on this OR allele are highest (**Figure 7F**). Thus, we can manipulate the zonal expression of an OR allele in mOSNs, by pharmacologically modulating the frequency and levels of transcriptional activation in INP/iOSN cells.

## Discussion

We uncovered a mechanism by which a probabilistic transcriptional process becomes skewed towards specific outcomes, transforming the relative position of a neuron across the dorsoventral axis of the MOE into biased OR gene choice. The solution to the perplexing segmentation of the MOE into distinct and reproducible territories of OR expression may be the following: polygenic OR transcription in neuronal progenitors highlights a small group of ORs that can be chosen for singular expression later in development (**Figure 7G**). In each MOE segment this OR mixture includes ORs that should be expressed in mOSNs of the segment, as well as ORs that are only expressed in more dorsal MOE segments (**Figure 7G**). As these progenitor cells differentiate into iOSNs, heterochromatic silencing may preferentially decommission from this mixture more dorsal ORs, and with lower efficiency, ORs that could be expressed in the segment, biasing this singular choice towards a spatially appropriate OR repertoire (**Figure 7G**). Our scRNA-seq analysis revealed two vectors in the determination of the OR ensemble that is co-transcribed in each OSN progenitor: chance, as every OR combination is unique, and determinism, as the overall zonal identities of the co-transcribed OR mixtures are informed by the position of the progenitor cell along the dorsoventral axis. Similarly, analysis of OR gene compartments with Hi-C revealed that genomic silencing also follows skewed patterns, eliminating preferentially ORs with more dorsal expression signatures than ORs that could be expressed in each zone, while Dip-C suggests that OR compartmentalization also retains an element of skewed randomness. The final product of these opposing probabilistic 'rheostats' may be the generation of gene expression programs that do not have sufficient resolution to determine which specific OR will be chosen in every OSN but are precise enough to generate reproducible dorsoventral expression territories for each one of the ~1400 OR genes. Recent reports describing opposing effects of chromatin compaction and transcriptional activation in the probabilistic expression patterns of the Spineless gene in R7 photoreceptors argue for the generality of this regulatory principle in vertebrates and invertebrates (**Voortman et al., 2022**).

We identified gradients of transcription factors NFI A, B, and X as partial orchestrators of the dorso-ventral patterning of OR expression, which they establish as follows: they contribute to the silencing of some dorsomedial (primarily zone 3) ORs; they activate both polygenic transcription and silencing of ventral (zone 4 and 5) ORs; and they have no influence on the expression of dorsal-most (class I and zone 1) ORs. Without NFI transcription factors, the majority of the MOE, excluding dorsal-most zone 1, defaults to a zone 2 identity. Given that NFI factors are predominantly known as regulators of embryonic and adult stem cell biology (*Clark et al., 2019*; *Adam et al., 2020*), it is surprising that in the olfactory system, their deletion does not interfere with the maintenance of stem cell populations, but with the OR expression patterns in post-mitotic, fully differentiated mOSNs. Interestingly, triple NFI deletion after the onset of singular OR choice has no effect on OR patterning, consistent with the emerging model that OR specification takes place exclusively at the INP to iOSN transition, and the notion that these patterning factors are not required for maintenance of OR transcription. Thus, we speculate that singular OR gene choice in OSNs can be executed by the common nucleoprotein complex of Lhx2/Ebf/Ldb1 bound to the multi-enhancer hub, consistent with the fact that we detect hubs of similar constitution associating with active ORs in different zones (*Monahan et al., 2019*).

A question emerging from these observations is why not use the same transcription factor gradients to regulate both polygenic and monogenic OR transcription? The answer is likely related to the abso-lute requirement for transcriptional singularity: transcription factor gradients can transcribe specific OR mixtures in a DV-responsive fashion, but they cannot activate only a single OR promoter among the many they can bind to. But even if singularity was achievable by transcription factor combinations and the OR-elicited feedback, OR promoters with the strongest binding motifs would be consistently chosen first, excluding ORs with weaker promoters in a 'winner takes all' model. This would result in preferential choice of specific ORs, reduced diversity in OR representation, and a narrower sensory spectrum for the olfactory system. With the process revealed here, any OR promoter activated in INPs/iOSNs is probabilistically chosen for singular expression in OSNs. Thus, by segregating OR gene regulation into two stages, polygenic transcription in progenitor cells and singular choice in OSNs, the olfactory system can impose deterministic biases while assuring equitable receptor representa-tion. Of course, this system has limitations in preserving transcriptional equity: artificial transcriptional induction of an OR allele in OSN progenitors under the powerful tetO promoter bypasses these constraints and results in a biased choice of this allele in most mOSNs. This immediately suggests that *cis* OR regulatory elements are subject to selective pressure that preserves their weak transcriptional activation properties, explaining why robust OR transcription in mOSNs requires the assembly of interchromosomal multi-enhancer hubs.

In this note, zones may also have evolved to satisfy the requirement for distributed OR represen-tation: if dorsal-most ORs, which are detected in every OSN progenitor regardless of DV origin, have the most frequently activated promoters, then silencing them in more ventral MOE segments assures that other OR identities will also have the chance to be expressed. Consistent with this model is the observation that mutations on the Lhx2 or Ebf binding sites of the promoter of dorsal OR M71 result in less frequent and more ventral M71 expression patterns (*Rothman et al., 2005*). Thus, DV segmen-tation of the MOE may serve as a mechanism that prevents ORs with stronger differences in promoter strength from competing for singular expression, assuring that every OR is expressed at meaningful, for odor perception, frequencies. In addition, as our spatial transcriptomic data showed, zonal regu-lation assures that ORs are expressed in a reproducibly patterned fashion in the MOE. While in wild-type mice unbiased machine learning approaches identify at least five distinct OR expression patterns, in the triple NFI cKO mice these patterns become intermixed for all but zone 1 ORs. With recent observations arguing that individual mitral cells, the second-order neurons in the olfactory circuit, have patterned projections in the brain (*Chen et al., 2022*), non-random OR expression in the MOE may contribute to putative hardwired components of odor perception and valence (*Kobayakawa et al., 2007*).

## Polygenic OR transcription as the arbiter between OR gene silencing and OR gene choice

A peculiar feature of the OR gene family that had emerged from our past work is that OR gene silencing is highest in the very cells that express ORs (*Magklara et al., 2011*). Our zonal analysis further strength-ened this intriguing correlation, as both H3K9me3/H3K79me3 and genomic compartmentalization in

each MOE segment are strongest on OR groups that are transcriptionally active during OSN differentiation. A fascinating implication of this observation is that early OR transcription is the signal for both genomic silencing and singular choice. Although the former is only implied by the strong correlation between OR transcription in OSN progenitors and genomic silencing, the latter is experimentally supported by the striking observation that strong transcriptional induction of *Olfr17* at the INP/iOSN stage results in strong recruitment of the Greek Island hub, and stable choice of this OR allele in most mOSNs throughout the MOE. Such a mechanism of promoter choice influenced by spatially-determined early transcription could also explain the recent demonstration that clustered Pcdh choice, which is regulated by anti-sense transcription (*Canzio et al., 2019*), abides by spatial restrictions in the neocortex (*Lv et al., 2022*).

How could two fundamentally opposite gene expression outcomes be encoded on the same molecular feature? We propose that the timing and levels of transcriptional induction could be the arbiters between genomic silencing and singular choice. ORs that are transcribed first in the INP stage, when the Greek Island hub cannot yet form due to the continuous expression of Lamin b receptor (*Clowney et al., 2012*), are most likely to be silenced. OR alleles activated during the assembly of the multi-enhancer hub, at the INP to iOSN transition, may compete for hub recruitment. The OR allele that will first associate with a multi-enhancer hub will be stably protected from heterochromatic silencing, possibly due to the significantly increased rates of OR transcription, whereas the other co-transcribed ORs will succumb to heterochromatic silencing. If timing and rates of OR transcription determine whether an OR allele will be silenced or chosen, then an OR allele that is highly transcribed in both INP and iOSN stages should evade silencing and dominate the competition for hub recruitment, explaining the striking expression pattern of the tTA-induced *Olfr17* allele. Thus, according to this model, in each OSN ORs with more dorsal identity will be transcribed first, because they have stronger promoters, and therefore will become silenced in higher frequency; ORs with the correct zonal identity will be transcribed later, with a chance to associate with the Greek Island hub, explaining why one is chosen and the rest are silenced; ORs with more ventral identities will not be transcribed at all, thus, will not be silenced but also will not be chosen. In other words, singular OR transcription may not depend on the silencing of every single OR in the genome: by encoding silencing and stable choice with the same exact molecular feature, OSNs choose one and silence a small fraction of the whole OR repertoire in each nucleus—the rest are not relevant. Notably, this constitutes a refinement of our original model, which proposed that all but one OR allele become silenced in each OSN (*Magklara et al., 2011*). Without knowledge of the zonal identity of most ORs, and without the technical ability to perform Hi-C and ChIP-seq on micro-dissected MOE zones, the striking patterns of gradual increase of OR silencing along the D-V axis of the MOE were not appreciated. As the cellular and temporal resolution of our genome-wide approaches increases, the aforementioned model will be further refined.

## Limitations of this study

Our experiments did not clarify whether NFI proteins bind directly on OR promoters, or act indirectly by activating other transcription factors and chromatin modifying enzymes. Although there is a statistically significant enrichment of NFI motifs on zone 4/5 OR promoters compared to the other OR promoters (data not shown), we were not able to detect direct binding of NFI proteins on these promoters, which is expected since these promoters are active in less than 1% of the cells. Given that our studies provide the mechanism by which NFI gradients establish zonal boundaries, via polygenic OR transcription and chromatin-mediated silencing, answering this question is not essential for understanding the mechanism of dorsoventral patterning of OR expression. A second limitation of this study is that it did not reveal the mechanisms that regulate the expression of the dorsal-most ORs (zone 1 ORs), as NFI deletion had no effects on the expression and chromatin regulation of these OR genes. However, having revealed the regulatory logic whereby these patterns are established, we expect that other transcription factors with zonal expression patterns identified here regulate early transcription and silencing of these genes across the MOE.

## Methods

### Experimental model and subject details

Mice were treated in compliance with the rules and regulations of IACUC under protocol numbers AC-AAAT2450 and AC-AABG6553. Mice were sacrificed using $CO_2$ following cervical dislocation. A complete list of mouse genotypes used for every experiment is in the *Supplementary file 2*. *Mash1-CreER* (also known as *Ascl1*$^{CreERT2}$) (*Kim et al., 2011*); *Ngn1-GFP* (*Magklara et al., 2011*) and Cre inducible tdTomato reporter (also known as *B6N.129S6-Gt(ROSA)26Sor*$^{tm1(CAG-tdTomato*,-EGFP*)Ees/J}$) (*Madisen et al., 2010*) mice were used to isolate four cell types in the olfactory lineage (GBC: tdTomato + GFP-, INP: tdTomato + GFP +, iOSN: tdTomato- GFP+ (bright), and mOSN: tdTomato + GFP dim) by sorting cells 48 hr after tamoxifen injection. We used young pups ranging from P2 to P4 at the start of the tamoxifen injection. GFP bright and dim cells from *Ngn1-GFP* pups (P6) were also used to isolate a mix of INP/iOSN cells and mOSN cells, respectively. *Omp-IRES-GFP* (*Shykind et al., 2004*) mice were used to isolate mature OSNs from adult (>8-week-old) mice. In order to obtain zonal iOSNs and mOSNs, *Olfr1507-IRES-Cre* (*Shykind et al., 2004*) and tdTomato alleles were crossed in with either *Ngn1-GFP* or *Omp-IRES-GFP* alleles to aid in zonal dissection (by labeling Ollfr1507 + expressing cells in zone 5).

Early knockout of NFI A, B, and X (NFI ABX) in horizontal basal cells (HBSs: the stem cell of the olfactory epithelium) was achieved by crossing *Nfia* fl/fl *Nfib* fl/fl and *Nfix* fl/fl triple conditional alleles, described in *Clark et al., 2019*, with *Krt5-CreER* (*Rock et al., 2009*) and tdTomato. Adult mice (>8-week-old) had deletion of NFI ABX in horizontal basal cells induced with three intraperitoneal injections with tamoxifen (24 hr apart). Ten days after the first injection, the olfactory epithelium was ablated with one intraperitoneal injection of methimazole, inducing proliferation of the HBCs and regeneration of a NFI ABX knockout olfactory epithelium. The olfactory epithelium was allowed to regenerate for 40 days before collecting the MOE and FAC-sorting the tdTomato+(dim) cell population, which contains a mixture of mostly mOSNs and some INP and iOSN cells, as described in detail in *Monahan et al., 2019*. For some experiments, *Omp-IRES-GFP* was crossed in to ensure all cells collected were mOSNs. To collect knockout INP cells the olfactory epithelium was only allowed to regenerate for 8–10 days before collecting the MOE and FAC-sorting the tdTomato+(dim) cells. We allowed the MOE to regenerate for 8 days for the RNA-seq experiments and 10 days for the native ChIP experiments, as heterochromatin deposition was still too low after 8 days to meaningfully analyze the pattern, although it followed the same trend as on day 10 (data not shown). As controls for all experiments (including RNA-seq, native ChIP-seq, Hi-C, imaging and spatial transcriptomics) involving early knockout of NFI ABX we used *Krt5-CreER*; tdTomato mice (in some cases with *Omp-IRES-GFP*). These control animals were age and sex matched and underwent the same tamoxifen induction and methimazole ablation as the knockout animals. Late knockout of NFI ABX in mOSNs was achieved by crossing *Nfia*, *Nfib*, and *Nfix* triple fl/fl alleles with tdTomato and *Omp-IRES-Cre*, and FAC-sorting tdTomato + cells from adult mice. Age and sex-matched *Omp-IRES-Cre*; tdTomato mice were used as controls for the late NFI ABX knockout experiments. A complete list of all the mouse genotypes can be found in *Supplementary file 2*.

Induction of *Olfr17* was achieved by crossing *tetO-Olfr17-IRES-GFP* mice (*Fleischmann et al., 2013*) with *Gng8-tTA* mice (*Nguyen et al., 2007*). To assess the stability of *tetO-Olfr17* expression after induction, adult mice >8 weeks were placed on a diet containing high doxycycline—200 mg/kg (Bio Serv, S3888)—for 35 days. To achieve a lower level of *tetO-Olfr17* induction, *tetO-Olfr17-IRES-GFP* mice were crossed with *Gng8-tTA* mice while being kept on a low amount of doxycycline in water—1 ug/ml (Sigma Aldrich, D9891) (*Redelsperger et al., 2016*) Mice were kept on doxycycline water throughout gestation and postnatal life, until collecting the MOE for analysis from mice >6 weeks old. A concentration of doxycycline in water at 5 ug/ml produced a similar pattern of expression as 1 ug/ml, albeit with fewer cells choosing *tetO-Olfr17*; and a concentration of 50 ug/ml and greater fully suppressed all *tetO-Olfr17* induction (data not shown).

### Method details

#### Zonal OR gene annotation

OR genes were assigned a zonal annotation (referring to their native zone of expression) as determined in *Tan and Xie, 2018*. We generated bins from their continuous zonal annotation by rounding to the nearest integer. There are a total of 1011 ORs with known zonal annotation. Of these, 115 are

Class I ORs, of which nearly all are expressed in zone 1, and 896 are Class II ORs, of which 261 are expressed in zone 1, 283 in zone 2, 164 in zone 3, 144 in zone 4, and 44 in zone 5. To have accurate mapping of OR promoters and gene bodies in all high throughput analysis we used the OR transcriptome generated by *Ibarra-Soria et al., 2014*.

## Zonal dissection of the olfactory epithelium

We used the fluorescent signal in *Olfr545-delete-YFP* (*Bozza et al., 2009*) (zone 1 OR), *Olfr17-IRES-GFP* (*Shykind et al., 2004*) (zone 2 OR), and *Olfr1507-IRES-GFP* (*Shykind et al., 2004*) (zone 5 OR) mice to practice dissections of dorsal (zones 1) MOE, dorsomedial (zone 2–3) MOE, and ventral (zone 4–5) MOE, respectively. Upon obtaining an accurate understanding of the zonal boundaries in the MOE we performed zonal dissections without the use of these fiduciary markers. Accuracy of dissections was confirmed by RNA-seq. For some experiments, *Olfr1507-IRES-Cre* and tdTomato reporter were crossed in to assist with accurate ventral (zone 5) MOE dissection (see **Supplementary file 2**).

## Fluorescence-activated cell sorting

Cells were prepared for FAC-sorting as previously described in *Monahan et al., 2019* by dissociating olfactory epithelium tissue with papain for 40 min at 37 °C according to the Worthington Papain Dissociation System. Cells were washed 2 x with cold PBS before passing through a 40 um strainer. Live (DAPI-negative) fluorescent cells were collected for RNA-seq and native ChIP-seq. Alternatively, for Hi-C cells were fixed for 10 min in 1% formaldehyde in PBS at room temperature, quenched with glycine, and washed with cold PBS before sorting fluorescent cells. Alternatively, for Dip-C, cells were fixed in 2% formaldehyde in PBS at room temperature for 10 min, inactivated with 1% BSA, and washed with cold 1% BSA in PBS before sorting fluorescent cells. All cells were sorted on a BD Aria II.

## Single-cell RNA-seq in olfactory lineage cell types

*Mash1-CreER; tdTomato; Ngn1-GFP* pups (ages P2-P4) were injected with tamoxifen and olfactory epithelium was collected after 48 hr. The tissue was dissected into ventral (zone 3–5) and dorsal OE (zone 1-2) sections, from which GBC (tdTomato+, GFP-), INP (tdTomato+, GFP+), iOSN (tdTomato-, GFP + bright) and mOSN (tdTomato-, GFP dim) cells were sorted into 384 well plates (split between the cell types). Each well of the 384 well plates had unique cell and molecular barcodes. Library preparation and sequencing was performed in collaboration with the New York Genome Center (NYGC) using a TSO approach for library preparation and sequenced on HiSeq2500. Reads were aligned to the mm10 genome according to the Drop-seq (*Macosko et al., 2015*) pipeline (http://mccarrolllab. org/dropseq/), which uses STAR for alignment, and discarding multi mapped reads with Samtools -q 255. Aligned single cells had a median of 133,686 unique transcripts (UMIs) and 2331 genes per cell (detected with a threshold of at least 3UMI). The experiment was performed in a biological replicate, resulting in 764 cells, from which we discarded cells with less than 1000 genes and 20,000 UMIs, resulting in 669 cells. We further filtered for cells that contained less than 5% mitochondrial reads, resulting in 591 cells used for analysis. We used Seurat v3 to normalize counts and cluster single cells, resulting in 6 populations. Clusters were assigned a cell-type based on expression of known olfactory lineage markers. We used the default setting of genes expressed in at least three cells for clustering but changed it to 1 when looking at OR expression (since expression of any OR out of >1000 genes is a rare event). For all OR expression analysis, we used a threshold of 3UMI for an OR to be considered expressed.

## Bulk RNA-seq in olfactory lineage cell types

GBC, INP, iOSN, and mOSN were isolated from *Mash1-CreER; tdTomato; Ngn1-GFP* pups as described above with the tissue being dissected into a ventral (mostly zone 4–5), dorsal OE (mostly zone 1) and a central section (that is enriched for zone 2-3). The experiment was performed in biological replicate. RNA was extracted from FAC-sorted cells using Trizol and libraries were prepared with the Nugen NuQuant RNA-seq library system and sequenced 50PE on HiSeq2500 or 75PE NextSeq (and trimmed to 50 bp before aligning). Cutadapt was used to remove adapter sequences and reads were aligned to the mm10 genome with STAR. Samtools was used to select high mapping

quality reads (-q 30). Normalization, calculation of FPKM (which we converted to TMP), and differential expression analysis were performed in R with DEseq2. For all RNA-seq data p-values refer to adjusted p-value (padj), which corrects for multiple hypothesis testing using the Benjamini-Hochberg method.

To find zone 5 enriched transcription factors at each developmental stage we used DEseq2 to determine significantly differentially expressed transcription factors (from the Gene Ontology database annotation 'DNA binding transcription factor activity') between ventral and dorsal cells with a padj less than 0.05 and at least a twofold change in expression (see *Supplementary file 1*.) To get the most likely candidates driving zonal identity we further filtered the list for transcription factors with at least a threefold difference between dorsal and ventral cells, and an expression level of at least 15 TPM.

## Zonal vs non-zonal mOSN markers from olfactory lineage RNA-seq data

To find ventrally enriched mOSN markers, we used DEseq2 to find non-OR genes differentially expressed between ventral mOSNs and dorsal or dorsomedial mOSNs (tomato-, GFP dim cells) with padj less than 0.05, and at least a twofold change in expression, of which there were 208; and performed the inverse analysis to generate a list of dorsal or dorsomedial enriched mOSN markers, of which there were 141 genes. To find non-zonal mOSN markers, we made a list of significantly upregulated genes (with a padj less than 0.05, and a fold change greater two) in mOSNs (tomato-, GFP dim cells) across all zones compared to iOSNs (tomato-, GFP + bright cells) across all zones. We further filtered out genes that were significantly differentially expressed between ventral and dorsal or dorsomedial mOSNs and took the top 200 most significant genes.

## RNA-seq in ventral NFI knockout mOSNs

To look at gene expression changes resulting from NFI deletion in olfactory progenitors we used NFI ABX triple knockout (*Nfia*, *Nfib*, and *Nfix* fl/fl; *tdTomato; Omp-IRES-GFP; Krt5-CreER*), NFI AB double knockout (*Nfia*, *Nfib* fl/fl; *tdTomato; Omp-IRES-GFP*, *Krt5-CreER*), NFIX only knockout (*Nfix* fl/fl, *tdTomato*, *Omp-IRES-GFP*, *Krt5-CreER*) or wt control (*tdTomato*, *Omp-IRES-GFP*, *Krt5-CreER*) mice and followed the same induction protocol for early knockout (described above) for all knockout genotypes as well as wt control animals. After rebuilding the MOE from knockout progenitors, we dissected ventral (zone 5) MOE and FAC-sorted GFP + mOSNs. RNA was extracted from sorted cells using Trizol and RNA-seq libraries were prepared with Nugen Nuquant RNA-seq library prep kit and sequenced 75PE on Nextseq 550. Reads were aligned exactly as described for zonal olfactory lineage data and similarly DEseq2 was used to determine differentially expressed genes between the different knockout and wt cells. To determine if ventral mOSN, dorsal mOSN, and non-zonal mOSN markers change in ventral NFI knockout cells, we analyzed the expression differences of the genes in our marker lists.

## Spatial transcriptomics

Whole MOE from NFI ABX knockout (using the *Krt5-CreER*) and wt control mice (two mice for each genotype) were embedded in OCT and frozen on dry ice. 14 µm cryosections of tissue were mounted onto Visium Spatial Gene Expression slides (10 X Genomics) and kept at –80 °C prior to processing. Tissue sections were fixed in methanol, stained with Hematoxylin and Eosin y, and imaged using a Nikon Eclipse Ti2 inverted microscope. Barcoded cDNA libraries of tissue sections were generated using the Spatial Gene Expression Reagent Kit (10 X Genomics) according to the manufacturer's protocols. Libraries were sequenced on an Illumina NovaSeq instrument at the University of Chicago Genomics Core with the following runtype: 28 cycles (Read 1); 10 cycles (i7 index); 10 cycles (i5 index); 120 cycles (Read 2). Data were demultiplexed and processed using SpaceRanger v1.1.0. Reads were aligned to the mm10 2020 A reference mouse transcriptome (10 X Genomics) and the OR transcriptome generated by *Ibarra-Soria et al., 2014*. After confirming the similarity between the two biological replicates, we did an additional round of sequencing of libraries from teo sections from one NFI ABX knockout and one wt control mouse to obtain deep data sets. This deep data was used for analysis of OR gene expression.

## Spatial transcriptomics analysis

Analysis was performed in R using STUtility (*Bergenstråhle et al., 2020*). Spatial spots expressing fewer than two OR genes and three OR transcripts were removed prior to analysis. Expression data across replicate sections were normalized using SCTransform, filtered to include only OR genes, and integrated using Harmony (*Korsunsky et al., 2019*). PCA was performed using the first five principal components and spatial spots were grouped into five clusters for both NFI ABX knockout and wt samples. Heatmaps were generated using the top 20 highest expressed differentially expressed genes (DEGs) within each zone (Class I through zone 5) for the wt sample and kept the same for the heatmap of the NFI ABX knockout. The same top 20 DEGs for zone 1, zone 2, and zone 5 were averaged per spot and overlaid against the H&E histology image (*Figure 5C*). For zonal spot assignment (*Figure 5D*), spots were designated to the zone with the largest summed normalized counts for all genes in that zone.

## Native chromatin immunoprecipitation from FAC-sorted cells

Native chip was performed as described in detail (*Monahan et al., 2017*). Unless otherwise indicated all steps were carried out at 4 °C. Briefly, FAC-sorted cells were pelleted at 600 rcf for 10 min in a swinging bucket centrifuge at 4 °C and resuspended in cold Buffer I (0.3 M Sucrose, 60 mM KCl, 15 mM NaCl, 5 mM $MgCl_2$, 0.1 mM EGTA, 15 mM Tris-HCl pH 7.5, 0.1 mM PMSF, 0.5 mM DTT, 1 x protease inhibitors). Cells were lysed by adding equal volume cold BufferII (Buffer I with 0.4% NP40) and incubating for 10 min on ice. Nuclei were pelleted 10 min at 1000 rcf and resuspended in 250 ul cold MNase buffer (0.32 M Sucrose, 4 mM $MgCl_2$, 1 mM $CaCl_2$, 50 mM Tris-HCl pH 7.5, 0.1 mM PMSF, 1 x protease inhibitors). Micrococcal Nuclease digestion was carried out by adding 0.1 U Micrococcal Nuclease (Sigma) per 100 ul buffer and incubating for 1 min 40 s in a 37 °C water bath, then stopping the digestion by adding EDTA to a final concentration of 20 mM. The first soluble chromatin fraction (S1) was collected by pelleting nuclei 10 min at 10,000 rcf at 4 °C and taking the supernatant to store at 4 °C overnight. Undigested, pelleted material was resuspended in 250 ul cold Dialysis Buffer (1 mM Tris-HCl pH 7.5, 0.2 mM EDTA, 0.1 mM PMSF, 1 x protease inhibitors) and rotating overnight at 4 °C. The second soluble chromatin fraction (S2) was collected by pelleting insoluble material 10 min at 10,000 rcf at 4 °C and taking the supernatant. S1 and S2 chromatin fractions were combined and used for immunoprecipitation with 5% material being retained for input. Equal cell numbers were used for control and knockout IPs, or between different cell types or zones. To perform IP, chromatin was diluted to 1 ml in Wash Buffer1 (50 mM Tris-HCl pH 7.5, 10 mM EDTA, 125 mM NaCl, 0.1% Tween-20, 5 mM 1 x protease inhibitors) and rotated overnight at 4 °C with 1 ug antibody. Dynabeads (10 ul Protein A and 10 ul Protein G per IP) were blocked overnight with 2 mg/ml yeast tRNA and 2 mg/mL BSA in Wash Buffer 1. Blocked beads were washed once with Wash Buffer 1, then added to antibody-bound chromatin and rotated 2–3 hr at 4 °C. Chromatin-bound beads were washed 4 x with Wash Buffer 1, 3 x with Wash Buffer 2 (50mM Tris-HCl pH 7.5, 10 mM EDTA, 175 mM NaCl, 0.1% NP40, 1 x protease inhibitors), and 1 x in TE pH 7.5. IP'd DNA was eluted by resuspending beads in 100 uL Native ChIP Elution Buffer (10 mM Tris-HCl pH7.5, 1 mM EDTA, 1% SDS, 0.1 M $NaHCO_3$) in a thermomixer set to 37 °C and 900 rpm for 15 min, repeating the elution 2 x and combining the eluates. IPs and inputs (diluted to 200 ul in elution buffer) were cleaned up with Zymo ChIP DNA columns (Zymo Research, D5205). Libraries were prepared with NuGEN Ovation V2 DNA-Seq Library Preparation Kit, and sequenced 50PE on HiSeq2500 and or 75PE on NextSeq 550.

## Native ChIP-seq analysis

Sequenced reads were pre-processed by trimming adapters with Cutadapt, then aligned to the mm10 genome using Bowtie2, with a default setting except for maximum insert size set to 1000 (-X 1000), allowing larger fragments to be mapped. Duplicate reads were removed with Picard, and high mapping quality reads were selected with Samtools (-q 30). After confirming the nChIP replicates looked similar, they were merged with HOMER and used to generate signal tracks at 1 bp resolution normalized to a library size of 10,000,000 reads. Thus, one of three replicates of H3K79me3 native ChIP-seq in zonal mOSNs was excluded for poor signal-to-noise ratio. Signal density over OR genes was calculated with HOMER annotatePeaks.pl then normalized to the length of each OR gene. Native ChIP heatmaps were generated with deeptools with OR gene bodies re-scaled to 6 kb and showing 2 kb flanking on each side.

## In situ Hi-C

In situ Hi-C and library preparation were performed exactly as described *Monahan et al., 2019*. Briefly, FAC-sorted cells (inputs ranged from 150,000–500,000 cells) were pelleted at 500 rcf for 10 min and lysed in Lysis buffer (50 mM Tris pH 7.5 0.5% NP40, 0.25% sodium deoxycholate 0.1% SDS, 150 mM NaCl, and 1 x protease inhibitors) by rotating for 20 min at 4 °C. Nuclei were pelleted at 2500 rcf, permeabilized in 0.05% SDS for 20 min at 62 °C, then quenched in 1.1% Triton-X100 for 10 min at 37 °C. Nuclei were then digested with DpnII (6 U/ul) in 1 × DpnII buffer overnight at 37 °C. In the morning, nuclei were pelleted at 2500 g for 5 min and buffers and fresh DpnII enzymes were replenished to their original concentration, and nuclei were digested for two additional hours. Restriction enzyme was inactivated by incubating for 20 min at 62 °C. Digested ends were filled in for 1.5 hr at 37 °C using biotinylated dGTP. Ligation was performed for 4 hr at room temperature with rotation. Nuclei were pelleted and sonicated in 10 mM Tris pH 7.5, 1 mM EDTA, 0.25% SDS on a Covaris S220 (16 min, 2% duty cycle, 105 intensity, Power 1.8–1.85 W, 200 cycles per burst, max temperature 6 °C). DNA was reverse crosslinked with RNAseA and Proteinase K overnight at 65 °C then purified with 2 × Ampure beads following the standard protocol and eluted in water. Biotinylated fragments were enriched with Dynabeads MyOne Strepavidin T1 beads and on-bead library preparation was carried out with NuGEN Ovation V2 DNA-Seq Library Preparation Kit, with some modifications: instead of heat inactivation following end repair beads were washed 2 x for 2 min at 55 °C with Tween Washing Buffer (TWB) (0.05% Tween, 1 M NaCl in TE pH 7.5) and 2 x with 10 mM Tris pH 7.5 to remove excess detergent. After ligation of adapters beads were washed 5 x with TWB and 2 x with 10 mM Tris pH 7.5. Libraries were amplified for 10 cycles and cleaned up with 0.8 V Ampure beads. Each experiment was performed with two biological replicates and prepared Hi-C libraries were sequenced 75PE on NextSeq 500.

## In situ Hi-C analysis

Reads were aligned to the mm10 genome using the distiller pipeline (*Goloborodko et al., 2022*; https://github.com/open2c/distiller-nf; *Open Chromosome Collective, 2023*), uniquely mapped reads (mapq >30) were retained and duplicate reads discarded. Contacts were then binned into matrices using a cooler. (*Abdennur and Mirny, 2020*). Analysis was performed on data pooled from two biological replicates, after confirming that the results of analysis of individual replicates were similar. Hi-C contact maps of OR clusters on chromosome 2 were generated with raw counts of Hi-C contacts normalized to counts/billion at 100 kb resolution. The maximum value on the color scale was set to 150 contacts per 100 kb bin. Analysis of zonal OR gene cluster contacts was performed with normalized counts binned at 50 kb resolution. All analyses were repeated using balanced counts generated by cooler (-mad-max 7), with similar results except balanced matrices discarded almost 10% of OR cluster bins due to relatively poor sequencing coverage.

## Dip-C

To isolate mature olfactory sensory neurons (mOSNs), Castaneous (Cas) mice were crossed to *Omp-IRES-GPF* mice. MOE was collected from adult heterozygous mice resulting from this cross. The tissue was dissected into zone 1 and zone 4/5, fixed for 10 min in 2% formaldehyde and FAC-sorted to isolate GFP + mOSNs. Dip-C was performed as described (*Tan et al., 2019*) on 96 mature OSNs: 48 each from dorsal and ventral MOE. Briefly, cells were lysed in Hi-C Lysis Buffer (10 mM Tris pH8, 10 mM NaCl, 0.2% NP40, 1 x protease inhibitors) on ice for 15 min, nuclei were pelleted at 2500 rcf for 5 min at 4 °C, then resuspended in 0.5% SDS and permeabilized 10 min at 62 °C then quenched in 1.1% Triton X-100 15 min at 37 °C. Nuclei were digested in 1 x DpnII buffer and 6 U/ul DpnII enzyme and digested overnight at 37 °C. Nuclei were then washed once in Ligation Buffer, and resuspended in Ligation buffer with 10U T4 DNA Ligase (Life Tech), and incubated for 4 hr at 16 °C shaking at 600 rpm. After ligation nuclei were pelleted and resuspended in cold PBS with DAPI to a final concentration of 300 nM and GFP + cells were FAC-sorted into a 96 well plate with 2 ul lysis buffer (20 mM Tris pH 8, 20 mM NaCl, 0.15% Triton X-100, 25 mM DTT, 1 mM EDTA, 500 nM Carrier ssDNA, and 15 ug/mL Qiagen Protease) and lysed for 1 hr at 50 °C and inactivated 15 min at 70 °C. DNA was transposed by adding 8 ul transposition buffer (12.5 mM TAPS pH 8.5, 6.25 mM MgCl2, 10% PEG 8000) with ~0.0125 uLTn5 (Vanzyme) and incubated at 55 °C for 10 min, then stopped with transposome removal buffer (300 nM NaCl, 45 mM EDTA, 0.01% Triton X-100 with 100 ug/mL Qiagen Protease) and

incubated at 50 °C for 40 min and 70 °C for 20 min. Libraries were amplified 14 cycles with i5 and i7 Nextera primers, with unique barcodes for each cell. Libraries from all cells were pooled and cleaned up with Zymo DNA Clean and Concentrate Kit. Libraries were sequenced 150PE on NextSeq 550.

## Dip-C analysis

Sequenced Dip-C reads were processed according to the Dip-C pipeline (*Tan et al., 2018*; https://github.com/tanlongzhi/dip-c; *Tan, 2014*). Reads were aligned to mm10 with BWA mem, and hickit was used to determine the haplotype of each contact based on SNPs between Cas and *Omp-IRES-GPF* mice and make a model of the 3D genome. Since *Omp-IRES-GPF* mice were a mixture of Bl6/129 strains, we only included SNPs that were unique to Cas mice to distinguish homologs. After alignment, cells were filtered using several quality control metrics described in *Tan et al., 2019*: We excluded cells that had less than 20,000 reads, cells that had a low contact-to-read ratio, and cells that had a high variability in 3D structure across computational replicates. Only 4 of 96 cells failed these metrics. Overall, the median number of contacts across cells was over 400,000. Computational analysis OR genes and Greek Island enhancers, including computing average contact densities and analysis of the 3D models, was performed using the Dip-C pipeline. Average contact densities between OR genes were calculated with 'dip-c ard.' Pairwise distances between OR genes from the 3D models were extracted with 'dip-c pd.' Heatmaps of pairwise distance were either ordered by genomic position or reordered using hierarchical clustering. To determine the size of OR gene aggregates, the number of OR genes within a specified radius of was calculated with 'network_around.py.' 3D models were visualized with PyMol and used to generate slices of the nucleus.

## Code availability

Scripts used for the analysis and generation of plots are publicly available at https://github.com/lisa-bash/Bashkirova_Zonal_OR_2023; (*Bashkirova, 2023*).

## Antibodies

Olfr17 antibodies were raised in rabbits against epitope RRIIHRTLGPQKL located at the C-terminus of the OR protein. Olfr1507 antibody was described in *Barnea et al., 2004*. The following antibodies were used for native ChIP: H3K79me3 (Abcam ab2621) and H3K9me3 (ab8898).

# Acknowledgements

The authors have no competing interests in this work. Mice were treated in compliance with the rules and regulations of IACUC under protocol numbers AC-AAAT2450 and AC-AABG6553. We thank members of the Lomvardas lab for critical discussions and input on the manuscript. We thank Drs. Tom Maniatis, Richard Axel, Gary Struhl, and Abbas Rizvi for critical comments and discussions. SL acknowledges support from the National Institutes of Health Common Fund 4D Nucleome Program (Grant 5U01DA040582), and the National Institute of Deafness and Communications Disorders (Grant 5R01DC018745). SL was also supported by the Roy Vagelos Professorship. RMG was supported by NYSTEM contracts C030133 and C30290GG. GB was supported by 5R01DC013561 (NIH). Work in the AF lab was supported by grants from the NIH (1U19NS112953-01) and the Robert J and Nancy D Carney Institute for Brain Science. Carney Institute computational resources used in this work were supported by the NIH Office of the Director grant S10OD025181. Data Availability: Sequencing data (RNA-seq, scRNA-seq, native ChIP-seq, Hi-C, Dip-C, and spatial transcriptomics) reported in this paper are publicly available at GEO under accession number GSE158730. This paper also makes use of published H3K9me3 mOSN native ChIP-seq data available at GEO under accession number GSE93570 and Olfr17 + mOSN Hi-C data available at the 4DN Data Portal under the accession number 4DNESNYBDSLY.

---

# Additional information

### Competing interests

---

Benjamin M Shykind: is an employee of Prevail Therapeutics, a wholly-owned subsidiary of Eli Lilly and Company. Bianca J Marlin: Reviewing editor, eLife. The other authors declare that no competing interests exist.

### Funding

| Funder | Grant reference number | Author |
|---|---|---|
| National Institute on Deafness and Other Communication Disorders | R01DC018745 | Stavros Lomvardas |
| Common Fund | U01DA052783 | Stavros Lomvardas |
| New York State Stem Cell Science | C030133 | Richard M Gronostajski |
| New York State Stem Cell Science | C30290GG | Richard M Gronostajski |
| The BRAIN Initiative | U19NS112953 | Alexander Fleischmann |
| National Institute on Deafness and Other Communication Disorders | R01DC013561 | Gilad Barnea |

The funders had no role in study design, data collection and interpretation, or the decision to submit the work for publication.

### Author contributions

Elizaveta V Bashkirova, Conceptualization, Data curation, Formal analysis, Validation, Investigation, Writing – original draft, Writing – review and editing; Nell Klimpert, Formal analysis, Investigation, Visualization, Methodology; Kevin Monahan, Ira Schieren, Beka Stecky, Bianca J Marlin, Alexander Fleischmann, Investigation; Christine E Campbell, Jason Osinski, Longzhi Tan, Gilad Barnea, Xiaoliang Sunney Xie, Ishmail Abdus-Saboor, Benjamin M Shykind, Richard M Gronostajski, Resources; Ariel Pourmorady, Visualization; Stavros Lomvardas, Conceptualization, Supervision, Funding acquisition, Investigation, Writing – original draft, Project administration, Writing – review and editing

### Author ORCIDs

Nell Klimpert http://orcid.org/0000-0002-6166-8026
Gilad Barnea https://orcid.org/0000-0001-6842-3454
Ishmail Abdus-Saboor https://orcid.org/0000-0003-2120-0063
Bianca J Marlin https://orcid.org/0000-0002-4275-7891
Alexander Fleischmann http://orcid.org/0000-0001-7956-9096
Stavros Lomvardas https://orcid.org/0000-0002-7668-3026

### Ethics

This study was performed in strict accordance with the Guides for the Care and Use of Laboratory Animals of the National Institutes of Health. All the animals were handled according to approve institutional animal care and use committee (IACUC) protocols AAC-AABG6533 and AAC-AAAT2450 of Columbia University .

Reviewer #1 (Public Review): https://doi.org/10.7554/eLife.87445.3.sa1
Reviewer #2 (Public Review): https://doi.org/10.7554/eLife.87445.3.sa2
Reviewer #3 (Public Review): https://doi.org/10.7554/eLife.87445.3.sa3
Author Response https://doi.org/10.7554/eLife.87445.3.sa4

## Additional files

### Supplementary files

• Supplementary file 1. Complete list of transcription factors significantly differentially expressed between dorsal and ventral cells at various stages of olfactory sensory neuron (OSN) differentiation. In the main *Figure 4* we only included transcription factors with a threefold difference between zone

1 and zone 4–5 but here we have a less stringent list, including transcription factors with twofold differential expression (related to *Figure 4*).

- Supplementary file 2. Complete list of mouse genotypes analyzed in this manuscript (related to methods).
- Supplementary file 3. List of reagents used for spatial transcriptomics (related to methods).
- MDAR checklist

## Data availability

Sequencing data (RNA-seq, scRNA-seq, native ChIP-seq, Hi-C, Dip-C and spatial transcriptomics) reported in this paper are publicly available at GEO under accession number GSE158730. This paper also makes use of published H3K9me3 mOSN native ChIP-seq data available at GEO under accession number GSE93570 and Olfr17+ mOSN Hi-C data available at the 4DN Data Portal under the accession number 4DNESNYBDSLY. Scripts used for analysis and generation of plots are publicly available at https://github.com/lisabash/Bashkirova_Zonal_OR_2023 copy archived at *Bashkirova, 2023*.

The following dataset was generated:

| Author(s) | Year | Dataset title | Dataset URL | Database and Identifier |
|---|---|---|---|---|
| Bashkirova E, Lomvardas S, Klimpert N, Fleischmann A | 2023 | NFI transcription factors regulate zonal olfactory receptor expression | https://www.ncbi.nlm.nih.gov/geo/query/acc.cgi?acc=GSE158730 | NCBI Gene Expression Omnibus, GSE158730 |

The following previously published datasets were used:

| Author(s) | Year | Dataset title | Dataset URL | Database and Identifier |
|---|---|---|---|---|
| Monahan K, Lombvardas S | 2017 | Chromatin State and Binding of Lhx2 and Ebf in Olfactory Sensory Neurons | https://www.ncbi.nlm.nih.gov/geo/query/acc.cgi?acc=GSE93570 | NCBI Gene Expression Omnibus, GSE93570 |
| Monahan K, Horta A, Lomvardas S | 2018 | HiC from mature olfactory sensory neurons expressing Olfr17 | https://data.4dnucleome.org/experiment-set-replicates/4DNESNYBDSLY/ | 4DN Data Portal, 4DNESNYBDSLY |

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
