## [Editor Report · eLife assessment]

This is an **important** paper that revises the canonical model of how olfactory sensory neurons choose which odor receptor to express. The data presented in the paper are **convincing** and the model proposed is provocative and likely to enable important future work.

---

## [Referee Report · Reviewer #1 (Public Review)]

Mature mammalian olfactory sensory neurons (OSN) express only one of the hundreds of possible odor receptors (ORs) encoded in the genome. The process of selecting this OR in each OSN is the consequence of both deterministic developmental processes involving transcription factors, and more stochastic processes. How this balance is implemented is a major problem in molecular neuroscience, one whose solution has significant systems-level implications for odor coding. In Bashkirova et al the authors substantially revise the canonical view of how this process works. By querying single cell transcriptomes and genetic architecture across OSN development, the authors demonstrate that OSN progenitors express ORs for their zone and for more dortsal zones, and that the degree of heterochromatinization of non-expressed ORs varies as a function of which zone a given OSN resides in. Through additional genetic experiments (including knockouts of transcription factors that seem to be associated with zonal identity, and the clever use of OR transgenes) they synthesize these findings into a model in which progenitors co-express many ORs - both ORs that are appropriate for their zone and ORs that are dorsal to their zone - and that this expression both facilitates heterochromatinzation of non-selected and extra-zonal ORs, and enables singular OR selection. The experiments are careful and the data are novel, and definitely revise our simplistic current view of how this process works; as such this work will have significant impact on the field. As presented the model requires additional experiments to fully flesh it out, and to definitively demonstrate that i.e., precocious expression leads to gene silencing, but with some additional clarifications in the discussion this paper both breaks new ground and sets the stage for future work exploring mechanisms of OSN development and OR selection.

---

## [Referee Report · Reviewer #2 (Public Review)]

In this study, Bashkirova et al. analyzed how the gene choice of olfactory receptors (ORs) is regulated in olfactory sensory neurons (OSNs) during development. In the mouse olfactory system, there are more than 1000 functional OR genes and several hundred pseudogenes. It is well-established that each individual OSN expresses only one functional OR gene in a mono-allelic manner. This is referred to as the one neuron - one receptor rule. It is also known that OR gene choice is not entirely stochastic but restricted to a particular area or zone in the olfactory epithelium (OE) along the dorsoventral axis. It is interesting to study how this stochastic but biased gene-choice is regulated during OSN development, narrowing down the number of OR genes to be chosen to eventually achieve the monogenic OR expression in OSNs.

In the present study, the authors cell-sorted OSNs into three groups; immediate neuronal precursors (INPs), immature OSNs (iOSNs), and mature OSNs (mOSNs). They found that OR gene choice is differentially regulated positively by transcription factors in INPs and negatively by heterochromatin-mediated OR gene silencing in iOSNs. The authors propose that by the combination of two opposing forces of polygenic transcription (positive) and genomic silencing (negative), each OSN finally expresses only one OR gene out of over 2000 alleles in a stochastic but stereotypic manner.

The authors' model of OR gene choice is supported by well-designed experiments and by large amounts of data. In general, the paper is clearly written and easy to follow. It will attract a wide variety of readers in the fields of neuroscience, developmental biology, and immunology. The present finding will give new insight into our understanding of gene choice in the multigene family in the mammalian brain and shed light on the long-standing question of monogenic expression of OR genes.

---

## [Referee Report · Reviewer #3 (Public Review)]

This manuscript investigates how a seemingly random choice of odourant receptor (OR) gene expression is organised into sterotypic zones of OR expression along the olfactory epithelium. Using a varietty of functional genomics methods, the authors find that along the differentiation axis (progenitor to mature olfactory sensory neuron, OSN) multiple ORs are initally transcribed and from among these, only one OR is selected for expression. The rest are suppressed through chromatin silencing. In addition to this, the authors report a dorso-ventral gradient in OR expression at the immature stage - dorsally expressed ORs are also expressed ventrally and these then get silenced. The expression of the ventrally expressed ORs, on the other hand, are restricted to the ventral region. They suggest a role for the transcription factor NF1 in this dorsoventral process.

This is a valuable study. The data are compelling and generally well presented.

---

## [Author Response]

The following is the authors’ response to the original reviews.

We are grateful to the 3 reviewers and the editorial team for agreeing that our work is rigorous and valuable for the fields of olfaction and developmental biology. We provide a revised version of the manuscript that addresses major concerns raised by the reviewers and adheres to their suggestions.

Specifically:

-We clarify what is novel in this work and we cover the appropriate literature.

-We tone down the language and interpretation of our data

-We clarify the categorization of zones and improve the readability to the best of our ability.

We have also made every effort to address minor points raised by the 3 reviewers and made clarifications wherever requested.